# QES-Plume v1.0: A Lagrangian dispersion model

Fabien Margairaz[1], Balwinder Singh[2], Jeremy A. Gibbs[3], Loren Atwood[1], Eric R. Pardyjak[1], and Rob Stoll[1]

[1]University of Utah, Department of Mechanical Engineering, 1495 E 100 S, Salt Lake City, UT, USA
[2]Pacific Northwest National Laboratory, 902 Battelle Boulevard, Richland, WA, USA
[3]NOAA/OAR National Severe Storms Laboratory, Norman, OK, USA

**Correspondence:** Fabien Margairaz (fabien.margairaz@alumni.epfl.ch)

**Abstract.** Low-cost simulations providing accurate predictions of transport of airborne material in urban areas, vegetative canopies, and complex terrain are demanding because of the small-scale heterogeneity of the features influencing the mean flow and turbulence fields. Common models used to predict turbulent transport of passive scalars are based on the Lagrangian stochastic dispersion model. The Quick Environmental Simulation (QES) tool is a low-computational-cost framework developed to provide high-resolution wind and concentration fields in a variety of complex atmospheric-boundary-layer environments. Part of the framework, QES-Plume, is a Lagrangian dispersion code that uses a time-implicit integration scheme to solve the generalized Langevin equations which require mean flow and turbulence fields. Here, QES-plume is driven by QES-Winds, a 3D fast-response model that computes mass-consistent wind fields around buildings, vegetation, and hills using empirical parameterizations, and QES-Turb, a local mixing-length turbulence model. In this paper, the particle dispersion model is presented and validated against analytical solutions to examine QES-Plume's performance under idealized conditions. In particular, QES-Plume is evaluated against a classical Gaussian-plume model for an elevated continuous point-source release in uniform flow, the Lagrangian scaling of dispersion in isotropic turbulence, and a non-Gaussian-plume model for an elevated continuous point-source release in a power-law boundary-layer flow. In these cases, QES-plume yields a maximum relative error below $6\%$ when compared with analytical solutions. In addition, the model is tested against wind-tunnel data for a uniform array of cubical buildings. QES-Plume exhibits good agreement with the experiment with $99\%$ of matched zeros and $59\%$ of the predicted concentrations falling within a factor of 2 of the experimental concentrations. Furthermore, results also emphasize the importance of using high-quality turbulence models for particle dispersion in complex environments. Finally, QES-Plume demonstrates excellent computational performance.

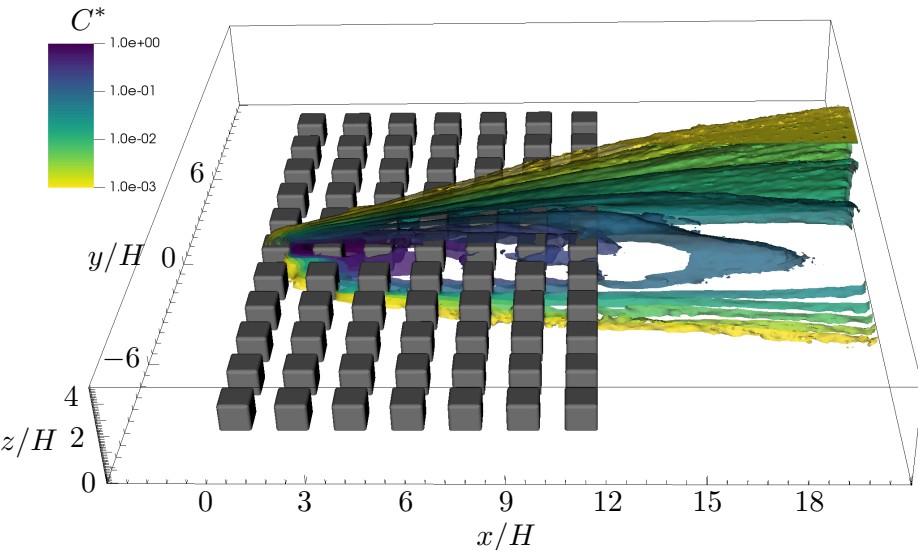

Key figure:

## 1 Introduction

Rapid growth of urban populations around the world impacts all sectors of human activity, including industrial and transportation. Additionally, growth is also increasing pressure on agricultural systems to boost yields. These trends raise concerns about deterioration of the environment, decline of quality of life, or worsening air quality (Britter and Hanna, 2003). Along with these long-term problems, acute risks including the potential accidental or deliberate release of a chemical or biological agent pose a major threat in densely populated urban areas (Gowardhan et al., 2021).

In response to these and other issues, a number of fast-response transport and dispersion models for urban areas and complex terrain have been developed. Fast-response models are characterized by their ability to keep computational costs low while providing realistic representations of the effects of buildings, canopies, and terrain on velocity distributions and the dispersion of scalars (Pardyjak et al., 2008; Singh et al., 2008; Gowardhan et al., 2011). Different classes of models exist for scalar dispersion. First, reduced-order models, like Gaussian dispersion models, use algebraic descriptions of plumes to calculate concentrations (Hanna et al., 2003; Philips et al., 2013; Prussin et al., 2015; Miller et al., 2018). Examples of operational reduced-order models include AERMOD (Cimorelli et al., 2005) or SIRANE (Soulhac et al., 2011). However, the performance of Gaussian models in complex environments is limited as this class of model struggles to capture aspects of the plume such as asymmetry, plume turning, or other critical features especially in urban settings (Hertwig et al., 2018; Pirhalla et al., 2021). Another class of models follows an Eulerian approach that solves the prognostic equation for concentration, such as the large-eddy-simulation suite PALM (Maronga et al., 2020), the hybrid Eulerian/Lagrangian Code Saturne (Archambeau

et al., 2004; Bahlali et al., 2018), and the CMAQ Modeling System (Byun and Schere, 2006; US EPA Office Of Research And Development, 2020). However, the sophistication and computational cost result in these models being unsuitable for rapid deployment at urban or local scales. The uses of these models is often more appropriate for research or at large, continental, or global scales. A third class, Lagrangian stochastic dispersion models (LSDMs), describe the movement of each particle in turbulent flows using a random-walk approach (Pope, 1987; Thomson, 1987; Rodean, 1996). Examples of operational LSDMs are QUIC-PLUME (Williams et al., 2002), MicroSpray (Tinarelli et al., 2013), and GRAMM/GRAL (Oettl, 2015). LSDMs have the benefit of being easily parallelized thanks to their mathematical formulation (Singh et al., 2011). In addition to the dispersion model, accurate estimations of concentrations of airborne substances require mean-wind and turbulence models such as QUIC-URB (Pardyjak and Brown, 2003), SIRANE (Soulhac et al., 2011), or street-network models (Hertwig et al., 2018). The importance of these models was emphasized by Carissimo et al. (2021), who evaluated how different numerical modelling approaches resolved the concentration in the wake of buildings.

Another area of study where fast-dispersion models can have a big impact is aerobiology, the study of aerial dispersion of biological particles in the atmosphere (Legg, 1983; Aylor, 2017). In particular, accurate prediction of temporal and spatial dispersion of pathogens can provide crucial information related to the development of plant disease epidemics or for fungicide resistance management (Mahaffee et al., 2014). An overview of the use of aerial sampling for detection of epidemic development can be found in Mahaffee et al. (2022). Similarly, dispersion models can be used to study long distance transport of pollen and spores (e.g., Aylor, 2003). In recent years, there has been an increase in interest from growers to adopt techniques from aerobiology to improve yields (Mahaffee et al., 2011; Thiessen et al., 2016).

Numerical integration of the equations used in LSDMs is challenging due to the stiffness of the model's mathematical formulation (Yee and Wilson, 2007). An equation is considered stiff when some numerical methods are unstable, unless extremely small time or space steps are used. In particular, for LSDMs a particle may travel large distances over small timesteps because of the existence of instabilities in the numerical methods. Simplified versions of the equations are not as unstable because of the reduced number of terms compared to the generalized model. However, simplified models are still numerically unstable. Furthermore, early numerical integration methods were known for violating the well-mixed condition, where uniformly distributed particles become unmixed even under homogeneous conditions because of numerical instabilities (Thomson, 1984). This phenomenon has been attributed to 'rogue' trajectories, where particles accumulate energy and develop arbitrarily large velocity fluctuations (Yee and Wilson, 2007; Postma et al., 2012; Wilson, 2013; Postma, 2015).

Several methods have been presented to integrate LSDMs. Yee and Wilson (2007) proposed a fractional step methodology to partially circumvent the stiffness in the generalized model. Ramli and Esler (2016) outlined a rigorous methodology to evaluate numerical schemes for LSDMs where a series of one-dimensional test problems were introduced based on the Fokker–Planck equation. They conclude that if long-timesteps have to be used–for long-distance transport, for example–it can be beneficial to use a random displacement model approximation rather than classical integration schemes. Bailey (2017) investigated the possibility of using a time-implicit scheme to eliminate rogue trajectories. He showed that numerical instabilities of the temporal integration scheme lead to nonphysical trajectories and that a lagged implicit scheme is unconditionally stable for the generalized model. Bailey's conclusion motivated the use of his methodology in the present work.

To address the issues presented above, the Quick Environmental Simulation (QES) framework was developed to provide high-resolution wind and concentration fields in complex urban and agricultural environments. The framework is composed of: QES-Winds, a 3D fast-response model that computes mass-consistent wind fields around buildings and vegetation using empirical parameterizations (Bozorgmehr et al., 2021; Margairaz et al., 2022b); QES-Turb, a turbulence model based on an eddy-viscosity parameterization with a local mixing length (Pope, 2000); and QES-Plume, a particle dispersion model. The objective of this work is to describe and validate QES-Plume.

In the following, we first describe the mathematical formulation of the LSDM in Sec. 2. Next, Sec. 3 introduces the QES framework and the results of the model validation are presented in Sec. 4. The importance of the wind flow and turbulence models is discussed in Sect. 5 and finally, conclusions are presented in Sec. 6.

## 2 Lagrangian stochastic dispersion model

The motion of passive tracers in turbulent flow can be described by a random-walk model (Thomson, 1987). The time-evolution of a fluid particle's position is given by

$$\frac{\mathrm{d}x_{\mathrm{p},i}}{\mathrm{d}t} = U_i + u_i, \tag{1}$$

where $x_{\mathrm{p},i}$ is the position of the particle p in a Cartesian coordinate system $i \in \{1,2,3\}$, $t$ is time, $U_i$ is the particle's mean velocity, and $u_i$ is its velocity fluctuation. Here, a 'particle' is a statistical representation of fluid element containing many molecules Pisso et al. (2019) and not an actual aerosol particle. The fluctuation is modelled by a random-walk process formalized in the Langevin equations (Langevin, 1909). This stochastic differential equation is given by

$$\mathrm{d}u_i = -au_i\mathrm{d}t + b_{ij}\mathrm{d}W_j, \tag{2}$$

where Einstein notation is used to imply summation over repeating indices. In Eq. (2), $a$ is a damping coefficient associated with viscous drag (Rodean, 1996) and $b_{ij}$ is a scaling tensor for the three independent random variates $\mathrm{d}W_j$ representing Brownian (or Wiener) processes (Thomson, 1987). The random terms $\mathrm{d}W_j$ follows a Gaussian distribution with zero mean and variance $\mathrm{d}t$ (Yee and Wilson, 2007).

For stationary, homogeneous, and isotropic turbulence, a simplified model is obtained by writing $a$ and $b_{ij}$ as a function of the Lagrangian velocity timescale $\tau_L$. In this case, the simplified Langevin equations (SLEs) are given by

$$\mathrm{d}u_i = -\frac{1}{\tau_L}u_i\mathrm{d}t + \left(\frac{2\sigma^2}{\tau_L}\right)^{1/2}\mathrm{d}W_i. \tag{3}$$

Following Rodean (1996), the timescale $\tau_L$ can be parameterized as

$$\tau_L = \frac{2\sigma^2}{C_0\bar{\varepsilon}}, \tag{4}$$

where $\sigma^2$ is the velocity variance, $C_0$ is a universal constant, and $\bar{\varepsilon}$ is the mean dissipation rate of turbulence kinetic energy (TKE). For $C_0$, Rodean (1991) reported values ranging from $1.6$ to $10$ with Du (1997) proposing a value of $C_0 = 3.0 \pm 0.5$ and for a constant-stress region, Rodean (1991) derived a semi-analytical value of $C_0 = 5.7$.

The model in Eq. (3) can be expanded for Gaussian, inhomogeneous, and anisotropic turbulence. Following Thomson (1987), the 3D generalized Langevin equations (GLEs) are given by

$$\mathrm{d}u_i = -\frac{C_0\bar{\varepsilon}}{2}\tau_{ik}^{-1}u_k\mathrm{d}t + \frac{\tau_{\ell j}^{-1}}{2}\frac{\mathrm{d}\tau_{i\ell}}{\mathrm{d}t}u_j\mathrm{d}t + \frac{1}{2}\frac{\partial\tau_{i\ell}}{\partial x_\ell}\mathrm{d}t + (C_0\bar{\varepsilon})^{1/2}\,\mathrm{d}W_i \tag{5}$$

where $\tau_{ij}$ is the Reynolds stress tensor and $\tau_{ij}^{-1}$ is its inverse. One of the consequences of the hypothesis of Gaussian turbulence is that the Reynolds stress tensor must be positive definite, otherwise the probability distribution function of the velocity fluctuations becomes ill-defined (Bailey, 2017). A tensor is positive definite if and only if all of its eigenvalues are positive. Since the GLEs require the existence of an inverse of $\tau_{ij}$, the modeled stress tensor has to be positive definite. Therefore, all eigenvalues and principal invariants of the stress tensor have to be positive, i.e., it must be realizable (Vachat, 1977).

The GLEs are considered stiff because of the presence of the wide range of time scales. In particular, a particle may travel a significant distance over a small timestep because of the existence of instabilities in the numerical solution. The SLEs drastically reduce the number of terms in the GLEs and are considered less unstable, however, numerical instabilities still exist for the SLEs (Yee and Wilson, 2007).

On the ground and at building surfaces, reflecting boundary conditions are used. At the domain top, an outlet condition is used, indicative of particles traveling past the top of the domain. Yet, the top boundary conditions can be more complex if the top of the atmospheric boundary layer is considered (see Thomson et al., 1997).

## 3  Method

To calculate the trajectory of fluid particles, the QES framework is composed of three main modules: (1) QES-Winds, a mass-consistent wind model; (2) QES-Turb, a turbulence model; and (3) QES-Plume, a LSDM. Figure 1 shows the workflow of the QES framework. First, the steady-state wind field is computed. The resulting velocity vector ($U_i$) is passed to the turbulence model, where the Reynolds stress tensor ($\tau_{ij}$) and the mean dissipation rate of TKE ($\bar{\varepsilon}$) are calculated. Finally, all variables are passed to QES-Plume. QES's modules are coded in C++ and Nvidia's CUDA application programming interface (Margairaz et al., 2022a). The input and output take advantage of the netCDF (Network Common Data Form) library developed by UCAR/Unidata (Rew et al., 1989). In addition, QES-Plume can also run as a stand-alone model with pre-computed velocity and turbulence fields. In this mode, QES-Plume is initialized directly from netCDF files. This mode is mainly used for verification and validation purposes.

### 3.1  QES-Winds: mass-consistent wind model

The wind field is obtained using QES-Winds. This model is based on the framework introduced by Sherman (1978) and Röckle (1989) to simulate a divergence-free steady-state 3D wind field around buildings. The framework uses a combination of a mass-consistent diagnostic wind model and various empirical parameterizations to describe the dynamics of the flow around buildings or through vegetated canopies. The mass-consistent solver employs a variational-analysis technique introduced by Sasaki (1970) where the Euler–Lagrange equation is used to minimize the error between the guess and final flow field under

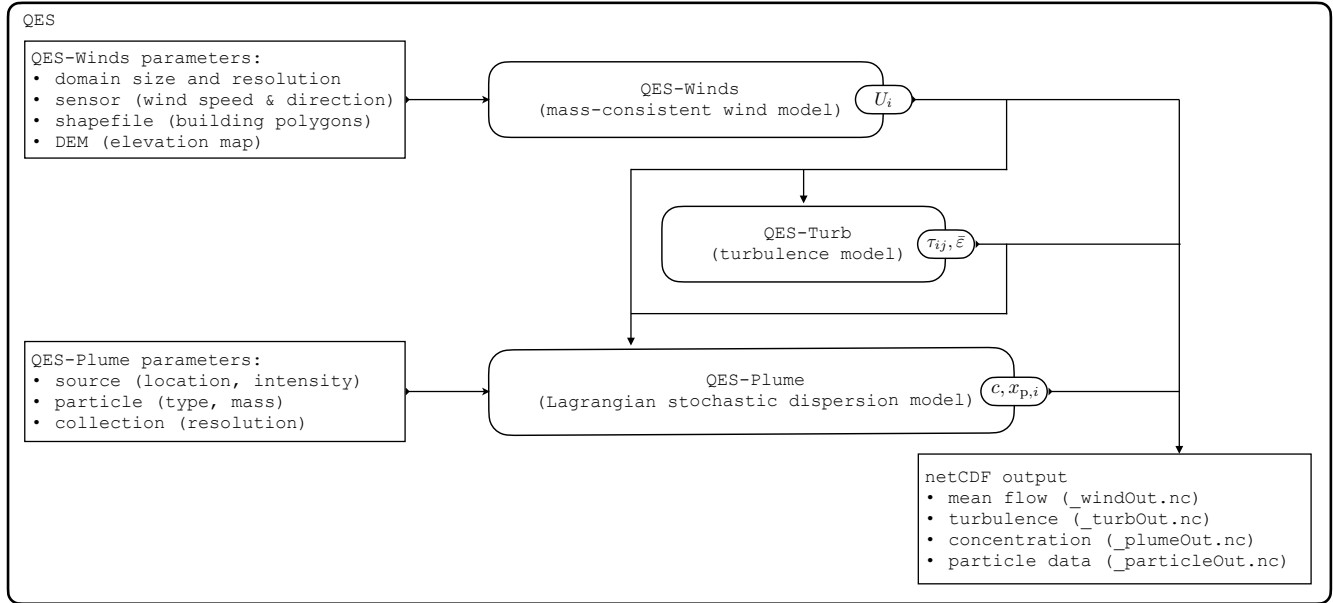

**Figure 1.** Simulation workflow of the QES framework.

the divergence-free constraint. The minimization process is achieved through the solution of a Poisson equation with a succes-

135 sive over-relaxation (SOR) solver parallelized on GPUs using a Red/Black method (Bozorgmehr et al., 2021). The impact of momentum transport on flow around buildings is parameterized using models inspired from the QUIC-URB model (Pardyjak and Brown, 2003), namely, the building rooftop recirculation or velocity attenuation, the upstream recirculation zone, and the downwind recirculation zone and velocity deficit wake (Röckle, 1989; Singh et al., 2008; Gowardhan et al., 2010; Brown et al., 2013). In addition, a street-canyon parameterization has been implemented based on Singh et al. (2008). Flow through

vegetated canopies is parameterized using bulk canopy attenuation (Pardyjak et al., 2008) and isolated-tree (Margairaz et al., 2022b) and row-organized canopy models (Ulmer et al., 2023).

### 3.2  QES-Turb: turbulence model

To accurately represent particle motion in turbulent flows, the LSDM needs the stress tensor and the dissipation rate of TKE. The QES framework calculates turbulence variables using a local-mixing model based on Prandtl's mixing-length and Boussi-

145 nesq eddy-viscosity hypotheses (Pope, 2000). The mixing-length model was selected following a fast-response philosophy, favoring low-cost algebraic models rather than models based on transport equations. From the turbulent-viscosity hypothesis, the Reynolds stresses($\tau_{ij} = \overline{u_i u_j}$ where $\overline{\circ}$ represents the time-averaging operator) are given by

$$\tau_{ij} = \frac{2}{3} k \delta_{ij} - 2\nu_T S_{ij}, \tag{6}$$

where $k = \frac{1}{2}\langle u_i u_i \rangle$ is the TKE, $\delta_{ij}$ is the Kronecker delta tensor, $\nu_T$ is the eddy viscosity, and $S_{ij}$ is the mean strain-rate tensor given by

$$S_{ij} = \frac{1}{2}\left(\frac{\partial U_i}{\partial x_j} + \frac{\partial U_j}{\partial x_i}\right). \tag{7}$$

The eddy viscosity is modeled as a product of a characteristic length scale and velocity scale. For boundary-layer flows, the classical approach is to calculate the eddy viscosity as

$$\nu_T = \ell_m^2 \left|\frac{\partial U}{\partial z}\right|, \tag{8}$$

with the mixing length specified as $\ell_m = \kappa z$ where $\kappa \approx 0.4$ is the von Kàrmàm constant and $z$ is the height above the ground. Smagorinsky (1963) proposed a generalization for the eddy viscosity calculated using the mean strain-rate tensor. This formulation is given by

$$\nu_T = \ell_m^2 \left(2\overline{S_{ij}S_{ij}}\right)^{1/2}. \tag{9}$$

For complex flows, such as flows around buildings, the mixing length is computed as $\ell_m = \kappa d_W$ where $d_W$ is the distance to the closest wall.

From the eddy-viscosity model, the TKE can be defined as

$$k = \left(\frac{\nu_T}{C_T \ell_m}\right)^2, \tag{10}$$

where $C_T$ is a model constant. In QES-Turb, this constant is set to $C_T = 0.55$, to obtain the correct behavior for the log-law region (Pope, 2000, chap. 10). Following the same philosophy, the mean dissipation rate $\bar{\varepsilon}$ scales as a velocity scale cubed then divided by a length scale. Hence, $\bar{\varepsilon}$ is modeled as

$$\bar{\varepsilon} = C_D^3 \frac{k^{3/2}}{\ell_m}, \tag{11}$$

where $C_D$ is a model constant. To be consistent with the definition of the TKE, this constant is related to $C_T$ such that $C_D = C_T$ (i.e., $C_D = 0.55$, Pope, 2000, chap. 10).

The strain-rate tensor from Eq. (7) is calculated from the local gradients of the steady-state velocity fields simulated by QES-Winds. The velocity gradients are computed using central finite differences away from the walls. At the walls, the derivatives are computed with first-order forward finite differences.

For boundary-layer flows, there is ample evidence suggesting that the normal stresses are anisotropic and that for flow aligned with the $x_1$ direction $\tau_{11} > \tau_{22} > \tau_{33}$. However, the stress formulations in Eq. (6) do not guarantee that the modeled stress emulates these observations. Hence, to ensure that the stress reflects the anisotropic condition, the following formulation is used:

$$\tau_{11}^{BL} = C_u^2 \tau_{11}, \tag{12}$$

$$\tau_{22}^{BL} = C_v^2 \tau_{22}, \text{ and} \tag{13}$$

$$\tau_{33}^{BL} = C_w^2 \tau_{33}, \tag{14}$$

**Table 1.** Summary of $C_{\{u,v,w\}}$ values from the literature for different types of flows ([†]value not reported).

| Flow type | $C_u = \sigma_u/u_*$ | $C_v = \sigma_v/u_*$ | $C_w = \sigma_w/u_*$ | Reference |
|---|---|---|---|---|
| Rural area | 2.5 | 1.9 | 1.25 | Table 4 in Roth (2000) |
| Urban area averages | 2.32–2.49 | 1.91–1.99 | 1.27–1.29 | Table 4 in Roth (2000) |
| Urban area | 2.4 | 1.9 | 1.3 | Britter and Hanna (2003) |
| Lower part of vegetated canopy | 0.75 | $\sim$[†] | 0.5 | Brunet (2020) |
| Top of vegetated canopy | 2.0 | $\sim$[†] | 1.0 | Finnigan (2000); Brunet (2020) |
| Above vegetated canopy | 2.5 | $\sim$[†] | 1.25 | Finnigan (2000); Brunet (2020) |
| Wind tunnel over flat terrain | 2.5 | 1.8 | 1.2 | Castelli et al. (2001) |

where the constants $C_{\{u,v,w\}}$ are defined as the velocity component standard deviations normalized by the friction velocity (i.e., $C_u = \sigma_u/u_*$, $C_v = \sigma_v/u_*$, and $C_w = \sigma_w/u_*$). Table 1 summarizes the value of these constants for different types of flows found in the literature for neutral stability. The current version of the turbulence model does not account for atmospheric stability, as this study is focused on the urban canopy sublayer, where the atmosphere is mostly neutral (Ramamurthy et al., 2007). Future versions should include corrections to these coefficients in the surface layer following Monin-Obukhov similarity theory (Stiperski and Calaf, 2023).

Because the turbulence model is a local-mixing model, it relies heavily on the magnitude of the local velocity gradients to estimate the stress tensor. This is problematic in regions where the velocity gradients are small and leads to the model predicting negligible turbulence, for example in the core of a street canyon. While multiple methods exist to address these issues (e.g. Williams et al. 2002), a non-local turbulence mixing coefficient $C_{nlm}$ is added to the diagonal elements of the stress tensor everywhere in the domain to enhance the mixing and account for processes such as advection of turbulence into regions with small gradients. The result of this addition is an increase in the magnitude of the turbulence while maintaining the influence of local velocity gradients in the stress tensor. The non-local mixing coefficient, $C_{nlm}$, may not be valid for general applicability as this may yield unreasonable results unless verified extensively with more experimental data. For the array of cubical buildings presented in this paper, this coefficient is set to $C_{nlm} \approx 0.3$ m$^2$ s$^{-2}$ based on background mixing from the wind-tunnel data (see Appendix A2).

### 3.3 QES-Plume: Lagrangian stochastic dispersion model

#### 3.3.1 Numerical implementation

A common approach is the use of an explicit scheme to integrate the GLEs. The scheme is conditionally stable with a condition on the timestep related to the velocity variance and mean dissipation rate. However, due to the stiffness of the equation, extremely small timesteps are required to maintain stability and numerical errors can inject more energy than the viscosity can dissipate. As a consequence, the particle velocity can become arbitrary large leading to rogue trajectories (Yee and Wilson, 2007). Bailey (2017) proposed using an implicit numerical scheme to eliminate the numerical instability. However, a fully

implicit scheme for the GLE is challenging because of forcing terms. In order to simplify the problem and avoid coupling between the position and Langevin equations, the forcing terms are evaluated at the current particle position. This approach, called "lagged" forcing, is used to avoid a costly iterative scheme. The implicit scheme for the velocity fluctuations from $t^{(n)}$ to $t^{(n+1)} = t^{(n)} + \Delta t$ is given by

$$u_i^{(n+1)} = u_i^{(n)} + \left[ -\left( \frac{C_0 \bar{\varepsilon}}{2} \tau_{ik}^{-1} \right)^{(n)} u_k^{(n+1)} + \left( \frac{\tau_{\ell j}^{-1}}{2} \frac{\Delta \tau_{i\ell}}{\Delta t} \right)^{(n)} u_j^{(n+1)} + \frac{1}{2} \left( \frac{\partial \tau_{i\ell}}{\partial x_\ell} \right)^{(n)} \right] \Delta t + \left( C_0 \bar{\varepsilon}^{(n)} \right)^{1/2} \Delta W_i \tag{15}$$

where $u_i^{(n)}$ is the fluctuation at time $t^{(n)}$ and $u_i^{(n+1)}$ is the fluctuation at time $t^{(n+1)}$. Note that all the forcing terms are lagged (i.e., evaluated at time $t^{(n)}$). In particular, the time derivative of the stress tensor is approximated by

$$\frac{\mathrm{d}\tau_{ij}}{\mathrm{d}t} \approx \left( \frac{\Delta \tau_{ij}}{\Delta t} \right)^{(n)} = \frac{\tau_{ij}^{(n)} - \tau_{i\ell}^{(n-1)}}{\Delta t}. \tag{16}$$

This scheme is unconditionally stable but the system of equations in (15) requires the solution of a 3×3 matrix. Similar to the observation made by Bailey (2017), the authors have never observed the matrix to become singular, provided that the forcing terms are well defined.

Finally, a forward Euler scheme is used to update the particle position. The position $x_i^{(n+1)}$ is given by

$$x_i^{(n+1)} = x_i^{(n)} + \left( U_i^{(n)} + u_i^{(n+1)} \right) \Delta t \tag{17}$$

where $U_i^{(n)}$ is the mean velocity at the position $x_i^{(n)}$.

The workflow of the QES-Plume model is presented in Fig. 2. At each timestep, all sources emit particles based on each source's parameters. Typically, the particle's initial fluctuations are obtained by interpolating the velocity variances at the source location. Then, the motion of all particles is computed in the advection routine. First, the value of the velocity, stress tensor, stress tensor divergence, and mean dissipation rate are interpolated at the particle position using a trilinear interpolation method. To ensure that the model is well defined, the stress tensor must be made realizable if it is not, which can be either because of the turbulence model or interpolation. The stress tensor is realizable if all all three of its principal invariants are strictly larger than a tolerance of $10^{-5}$ (Bailey, 2017). If this condition is not satisfied, the diagonal elements of the stress tensor are incremented by 5% of the mean TKE. This procedure is repeated until all three principal invariants are larger than the selected tolerance. Enforcing this condition guarantees that the stress tensor can be inverted. Next, the velocity fluctuations are computed by solving Eq. (15) following the procedure in Fig. 2. Here, both the stress tensor $\tau_{ij}$ and the matrix $A_{ij}$ are explicitly inverted using the adjugate method (Horn and Johnson, 2012). Finally, the position of the particle is updated using Eq. (17) and the boundary conditions are applied, such as a reflecting boundary condition at the ground or on building faces (see Sec. 3.3.3), or an outlet condition on the sides and top of the domain (where particles are deleted once they travel outside of the domain). Once all particle positions have been updated, the relevant statistics are computed, in particular, the airborne concentration, and the time is incremented.

The new implementation of the LSDM has been tested following the procedure proposed by Bailey (2017). These tests checked that the well-mixed condition is respected for a wide range of forcing conditions and timesteps. To verify that unmixing

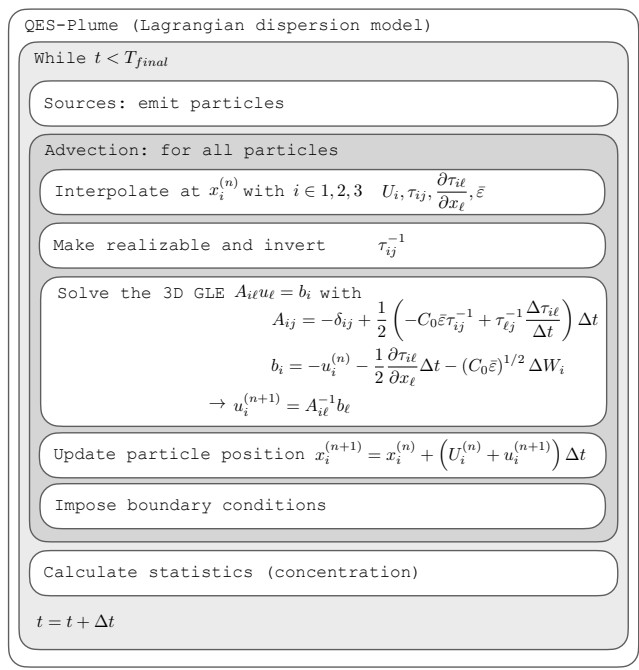

```
QES-Plume (Lagrangian dispersion model)

While t < T_final

    Sources: emit particles

    Advection: for all particles

        Interpolate at x_i^(n) with i ∈ 1,2,3    U_i, τ_ij, ∂τ_il/∂x_l, ε̄

        Make realizable and invert        τ_ij^-1

        Solve the 3D GLE A_il u_l = b_i with
                    A_ij = -δ_ij + 1/2 (-C_0 ε̄ τ_ij^-1 + τ_lj^-1 Δτ_il/Δt) Δt
                    b_i = -u_i^(n) - 1/2 ∂τ_il/∂x_l Δt - (C_0 ε̄)^(1/2) ΔW_i
                    → u_i^(n+1) = A_il^-1 b_l

        Update particle position x_i^(n+1) = x_i^(n) + (U_i^(n) + u_i^(n+1)) Δt

        Impose boundary conditions

    Calculate statistics (concentration)

    t = t + Δt
```

**Figure 2.** Workflow of the QES-Plume model using the implicit scheme to solve the 3D GLE. The particle position is advanced from its position $x_i^{(n)}$ at $t^{(n)}$ to its new position $x_i^{(n+1)}$ at $t^{(n+1)} = t^{(n)} + \Delta t$. The algorithm repeat the advection loop until the time reaches the final time $T_{final}$.

did not occur particles were uniformly distributed within the domain, the LSDM was run with different flow conditions and timesteps, and the final distribution of particle positions was checked. Three forcing conditions have been considered: ($i$)
synthetic data with zero flow and sinusoidal vertical stress, ($ii$) channel flow data from direct numerical simulation (Kim et al., 1987), and ($iii$) data from large-eddy simulation of the atmospheric boundary layer from Stoll and Porté-Agel (2006) (see Bailey, 2017 for more details about the tests).

### 3.3.2   Dynamic timestep

The method presented eliminates the possibility of the calculation of a rogue trajectory during particle advection. However,
some physical processes, such as reflection or deposition, require the particle to travel only one Eulerian-grid cell (i.e., QES-Winds velocity grid) at a time. To control the total distance travelled, a Courant-Number-based algorithm is used. The timestep is reduced as the particle moves close to a wall. The new timestep is calculated following the procedure presented in Algorithm 1. The user-specified timestep $\Delta t$ has to be progressively reduced as the particle moves closer to the wall using a user-specified Courant number $C_N$. The progressive reduction of the timestep is required to avoid carrying large fluctuations into a smaller
time increment. This procedure is used to divide the user-defined timestep into smaller time increments to ensure that the

**Algorithm 1** Courant-Number-based timestep reduction. $\Delta t$ is the user specified timestep. $C_N$ is the user specified Courant Number ($C_N < 1$).

$\Delta s = \min(\Delta x, \Delta y, \Delta z)$;
$U = \sqrt{(U_i^{(n)} + u_i^{(n)})(U_i^{(n)} + u_i^{(n)})}$;
$d$ distance to the closest wall;
**if** $d > 6U\Delta t$ **then**
    $\Delta t^{(n+1)} = \Delta t$;
**else**
    **if** $d > 3U\Delta t$ **then**
        $C = \min(2C_N, 1.0)$;
    **else**
        $C = C_N$
    **end if**
    $\Delta t^{(n+1)} = C\Delta s/U$;
**end if**

particle travels small distances. Hence, other algorithms requiring that the particle only travels one grid cell can be executed correctly.

### 3.3.3 Reflecting boundary condition

Perfect rebound is used as wall boundary conditions for the particles (Brzozowska, 2013; Bahlali et al., 2020). Figure 3 represents the trajectory of a particle with a double reflection in a corner. The wall-reflection method is presented in Algorithm 2. This method is called every time a particle crosses the fluid-solid interface.

**Algorithm 2** Wall reflection

**while** Particle in solid **do**
    Find closest face along trajectory
    Find intersection with face $\mathbf{p}$
    $\mathbf{d}$ is the portion of the trajectory beyond the intersection with the face
    $\mathbf{r} = 2(\mathbf{d} \cdot \hat{\mathbf{n}})\hat{\mathbf{n}}$
    $\mathbf{x}' = \mathbf{p} + \mathbf{r} + \mathbf{d}$
**end while**
$\mathbf{x}^{(n+1)} = \mathbf{x}'$

This specular rebound approach is the most common treatment of the wall boundary condition for LSDM. However, Bahlali et al. (2020) remarked that this approach does not account for momentum exchange in the wall-tangential directions. Following Dreeben and Pope (1997) and Bahlali et al. (2020), the proper treatment of the boundary condition requires precise evaluation of

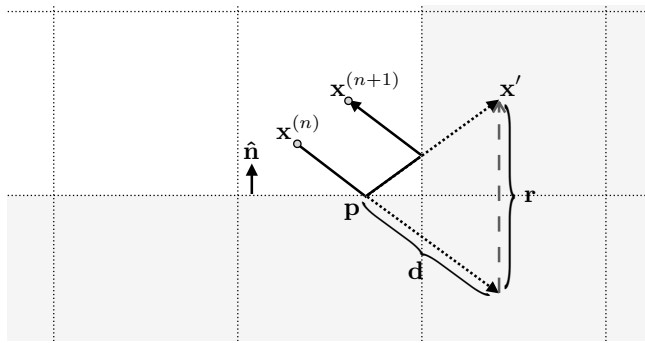

**Figure 3.** Example of double reflection in a corner from particle position $\mathbf{x}^{(n)}$ to $\mathbf{x}^{(n+1)}$. Dotted lines represent the part of the trajectory in the solid, represented by $\mathbf{d}$. Dash arrow represents the reflection $\mathbf{r} = 2(\mathbf{d} \cdot \hat{\mathbf{n}})\hat{\mathbf{n}}$, where $\hat{\mathbf{n}}$ is the wall normal. The particle position after first reflection is $\mathbf{x}' = \mathbf{p} + \mathbf{r} + \mathbf{d}$, where $\mathbf{p}$ is the intersection between the trajectory and the solid face.

the near-wall Reynolds stresses, which is not provided by the turbulence model. The reflection method can also be generalized for an arbitrary surface (Brzozowska, 2013).

### 3.3.4 Concentration

In this study, fluid particles are considered massless and therefore, the Eulerian concentrations are obtained by counting the number of particles $N(t)$ present in a sampling box of volume $V = \Delta x_s \Delta y_s \Delta z_s$ at each timestep during an averaging time $T$.
The average concentration is given by

$$C(x,y,z) = \frac{1}{VT} \sum N(t)\Delta t. \tag{18}$$

To compare concentration results between QES-Plume, analytical solutions, and experimental data, all concentrations are normalized using the following formula

$$C^* = C\frac{UH^2}{Q}, \tag{19}$$

where $C^*$ is the normalized concentration, $H$ is a reference height, often chosen as the height of the point of release, $U$ is a reference velocity, often chosen as the velocity at the point of release, and $Q$ is the source strength with units corresponding to the dimensional concentration.

## 4 Model evaluation

The performance of QES-Plume has been evaluated against two idealized test cases and a wind-tunnel test case for a $7 \times 11$
cubical array of buildings (Brown et al., 2001). The primary goal is to examine the acceptability of QES-Plume with the newly implemented GLE solver when compared to available analytical solutions and wind-tunnel data.

## 4.1 Continuous release in uniform flow

The normalized concentration profiles from the 3D-GLE model computations have been compared against classical Gaussian solution (Seinfeld and Pandis, 2015) for a steady-state, horizontally homogeneous, neutral atmospheric stability, constant wind speed and constant eddy diffusivity elevated continuous point source. The analytical concentration field is given by

$$C(x, y, z) = \frac{Q}{2\pi U \sigma_y \sigma_z} \exp\left(-\frac{y^2}{2\sigma_y^2}\right) \exp\left(-\frac{(z - z_s)^2}{2\sigma_z^2}\right), \tag{20}$$

where $Q$ is the source strength and $U$ is the wind speed at source height $z_s$. The plume standard deviations, $\sigma_y$ and $\sigma_z$, in the crosswind and vertical direction respectively, are given by $\sigma_y = \sigma_v F t$ and $\sigma_z = \sigma_w F t$ with $F = [1 + (t/T_i)^{1/2}]^{-1}$ and $T_i = (2.5 u_*/z_i)^{-1}$, where $t = x/U$ is the time of flight and $z_i$ is the boundary-layer height.

For this test case, the flow was prescribed by a horizontally and vertically uniform wind speed and friction velocity of $U = 2$ m s$^{-1}$ and $u_* = 0.174$ m s$^{-1}$, respectively. The boundary-layer height was set to $z_i = 1000$ m. In addition, the turbulence model had to be simplified for the horizontally homogeneous, constant eddy diffusivity and neutral stability conditions following Rodean (1996). The stress tensor was computed using the following algebraic expressions

$$\tau_{xx} = (2.5 u_*)^2 (1 - z/z_i)^{3/2}, \tag{21}$$

$$\tau_{yy} = (1.78 u_*)^2 (1 - z/z_i)^{3/2}, \tag{22}$$

$$\tau_{zz} = (1.27 u_*)^2 (1 - z/z_i)^{3/2}, \tag{23}$$

$$\tau_{xz} = -(u_*)^2 (1 - z/z_i)^{3/2}, \text{ and} \tag{24}$$

$$\tau_{xy} = \tau_{yz} = 0. \tag{25}$$

The mean dissipation rate of TKE was given by

$$\bar{\epsilon} = \frac{u_*^3}{\kappa z}(1 - 8.5 z/z_i)^{3/2}. \tag{26}$$

The particles were continuously released from a point source at height $H = 70$ m at a rate of 200 particles per second with a timestep of 1 second ($\Delta t = 1$ s) for a duration of 2100 s. To obtain statistically stationary concentration estimates, the concentration was averaged over 1800 s with a starting time of 300 s after the beginning of the release. The physical domain was broken up into 20, 49 and 69 sampling boxes in the $x$-, $y$- and $z$- directions, respectively, over a domain size of 100 m $\times$ 140 m $\times$ 140 m. The simulation parameters are summarized in Table 2.

Figure 4 shows the lateral and vertical normalized concentration profiles at three streamwise locations ($x/H = 0.464$, $x/H = 0.821$, and $x/H = 1.036$) for QES-Plume and the Gaussian analytical solution (Eq. (20)). The QES-Plume concentrations are in good agreement with the analytical solution with a coefficient of determination of $r^2 = 0.996$, a root-mean-square error (RMSE) for the normalized concentration of $0.165$ and a maximum relative error of $5.91\%$ over all profiles with $x/H \geq 0.2$. This threshold was chosen because the Gaussian plume model is known to be inaccurate close to the source (Stockie, 2011; Seinfeld and Pandis, 2015).

**Table 2.** Simulation parameters for continuous release in a horizontally and vertically uniform flow test case

| | |
|---|---|
| Domain size $(L_x, L_y, L_z)$ | 100 m, 140 m, 140 m |
| Uniform flow velocity $(U)$ | 2 m s$^{-1}$ |
| Friction velocity $(u_*)$ | 0.174 m s$^{-1}$ |
| Boundary layer height $(z_i)$ | 1000 m |
| LSDM constant $(C_0)$ | 5.7 |
| Advection timestep $(\Delta t)$ | 1 s |
| Concentration averaging time $(T)$ | 1800 s |
| Source location $(x_s, y_s, z_s = H)$ | 20 m, 70 m, 70 m |
| Source strength $(Q)$ | 200 particles/s |
| Sampling boxes $(\Delta x_s, \Delta y_s, \Delta z_s)$ | 5 m, 2 m, 2 m |

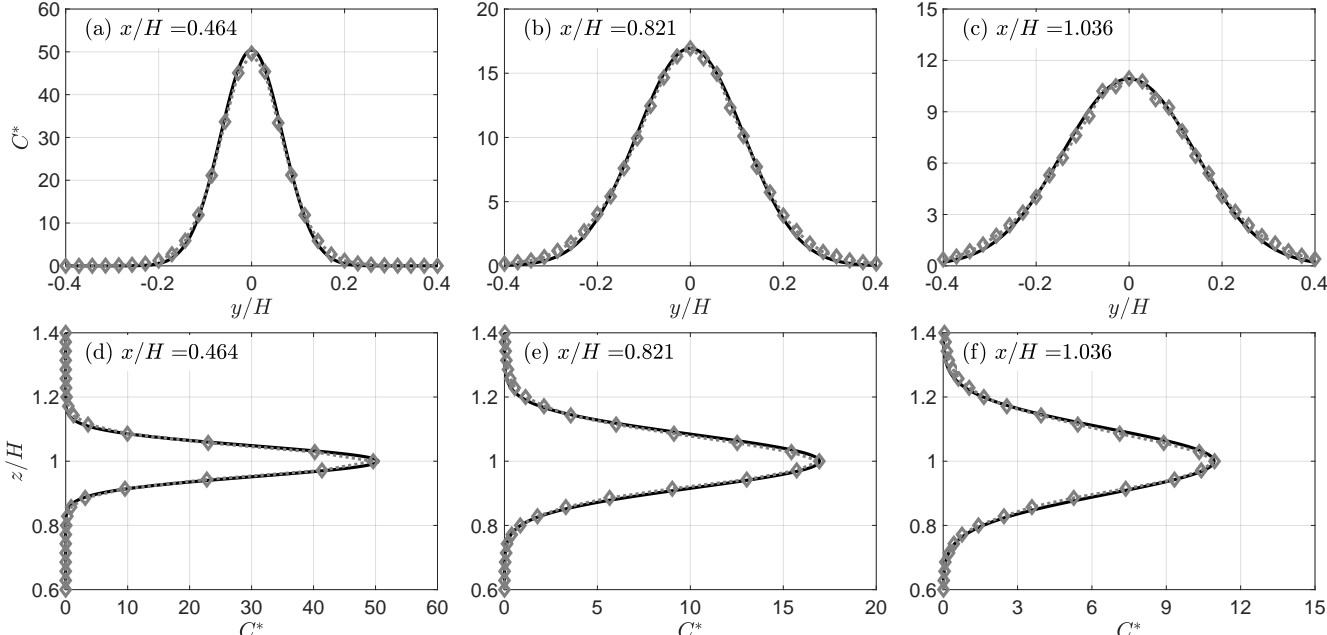

**Figure 4.** Profiles of the normalized concentration for the classical Gaussian plume model (line) and the QES-Plume model (diamond) at three different $x/H$ locations. The top panels are lateral profiles at $z/H = 1$, and bottom panels are vertical profiles at $y/H = 0$. Concentrations are normalized following Eq. (19).

In addition, this test case corresponds to the dispersion from a point source in statistically stationary isotropic turbulence presented in Pope (2000, chap. 12), following the Lagrangian approach introduced by Taylor (1921). To further validate QES-Plume, particle-trajectory statistics can be compared to this canonical example of turbulent dispersion. This approach shows that (i) trajectories close to the source consist essentially of straight-line motions, (ii) farther downstream the plume spread

adheres to the square-root of travel time, and (iii) the scaling variable is the Lagrangian timescale, $\tau_L$. To produce well-converged Lagrangian statistics, 100,000 particles were released from the source, the domain was extended to $L_x = 12$ km and $L_y = 8$ km, the source was placed at $y_S = 4$ km, and the total simulation time was increased to 7800 s.

The timescale, $\tau_L$, can be calculated based on the ratio of TKE to dissipation rate for the SLE, Eq. (4), with $\sigma^2 = 2/3\ k$. However, this method, which yields $\tau_L \approx 136$ s, is not valid for QES-Plume, which solves the 3D GLE. The timescale is computed using the autocorrelation function of the velocity fluctuation $\rho$. In this case, $\tau_L$ corresponds to the decorrelation timescale (or relaxation time) such that $\rho(s) \approx \exp(-s/\tau_L)$, where $s$ are the lags for the autocorrelation. Due to the limited length of the trajectories and large variability of the model, the autocorrelation function is averaged over all particles and the timescale is computed by fitting an exponential decay on lags smaller that 400 s. This method yields $\tau_L \approx 121$ s, which is very close the results from the SLE. According to Pope (2000, chap. 12), the standard deviation of the positions follows the approximated form given by

$$\sigma_Y(t) \approx \begin{cases} u't, & \text{for } t \ll \tau_L, \\ \sqrt{2u'^2\tau_L t}, & \text{for } t \gg \tau_L, \end{cases} \tag{27}$$

where $\sigma_Y$ is calculated on the spanwise position and $u'$ is the root-mean-square value of the velocity fluctuation from Eq. (5). Thus, the trajectories exhibit two distinct regimes with a region of linear spread and a region where spread follows a square root. Figure 5 presents the trajectories obtained in QES-Plume, for homogeneous isotropic turbulence from the uniform flow test case. The standard deviation of the spanwise position $\sigma_Y$ (Fig. 5a) matches both linear and square-root scaling regions. This behavior is visible in the sample of 200 trajectories which demonstrates the linear spread for $t < \tau_L$ (Fig. 5b) and the square-root spread for $t > \tau_L$ (Fig. 5c). Note that for $t/\tau_L > 50$, particles are exiting the domain, affecting the calculation of $\sigma_Y$, hence the degradation of the scaling for large times. In conclusion, QES-Plume reproduced the canonical dispersion behavior from the Langevin model for homogeneous isotropic turbulence (Pope, 2000).

## 4.2 Continuous release in a power-law atmospheric boundary-layer flow

The next test case examines the performance of the QES-Plume model against an existing analytical solution for a continuous point-source release in a boundary-layer flow. The source was relatively close to the ground ($H = 4$ m) to allow reflection of the emitted particles off of the ground. The QES-Plume normalized concentration profiles are compared against the classical non-Gaussian solution (Huang, 1979; Brown et al., 1993, 1997) for a steady state, horizontally homogeneous, neutral atmospheric stability, power-law wind profile $u(z) = az^p$ and power-law eddy diffusivity $K_z(z) = bz^n$. The analytical solution for the concentration is given by

$$C(x,y,z) = \frac{Q}{\sqrt{2\pi}\sigma_y} \exp\left[-\frac{y^2}{2\sigma_y^2}\right] \frac{(zz_s)^{(1-n)/2}}{b\alpha x} \exp\left[-\frac{a(z^\alpha + z_s^\alpha)}{b\alpha^2 x}\right] I_{-\nu}\left[\frac{2a(zz_s)^{\alpha/2}}{b\alpha^2 x}\right], \tag{28}$$

where $\alpha = 2 + p - n$, $\nu = (1-n)/\alpha$ and $I_{-\nu}$ is the modified Bessel function of first kind of order $-\nu$. The lateral standard deviation $\sigma_y$ is given by $\sigma_y = 0.32x^{0.78}$ (Seinfeld and Pandis, 2015). In this test case, the turbulence stress gradients were

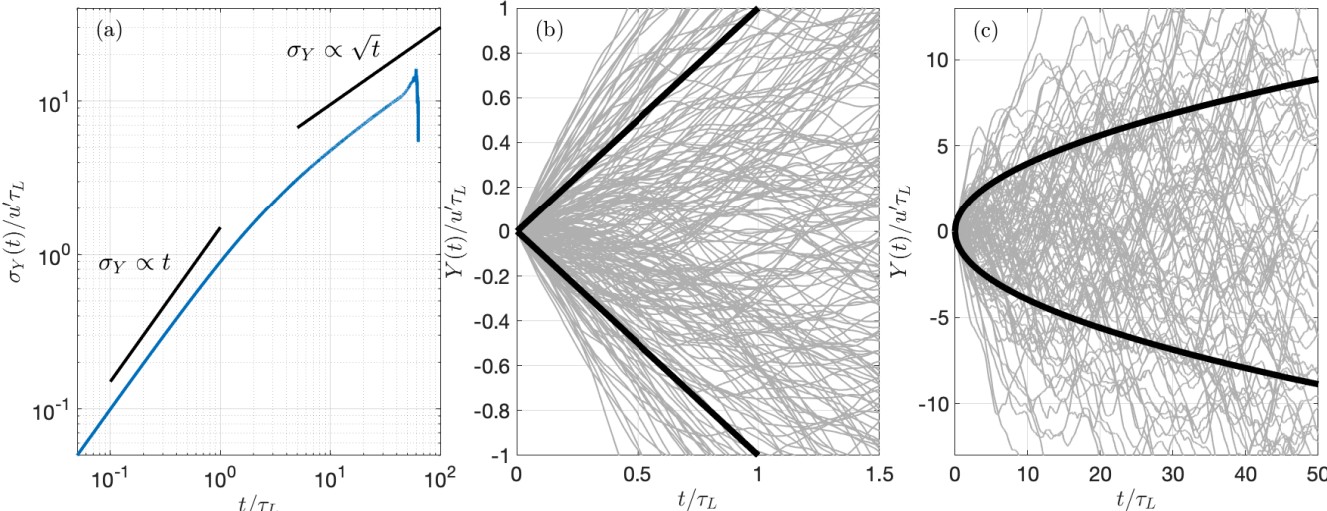

**Figure 5.** Scaling of trajectories in the spanwise direction for homogeneous isotropic turbulence showing the linear spread for $t < \tau_L$ and the square-root spread for $t > \tau_L$. Panel (a) shows the scaling of the standard deviation $\sigma_Y(t)$ of the horizontal spread. Panels (b) and (c) show a sample of 200 trajectories in the region of linear spread and square-root spread, respectively, where the black lines represent $\pm\sigma_Y$ from Eq. (27).

present due to the velocity gradients in the vertical direction. The friction velocity is given by $u_* = \kappa p a z^p$. The stress tensor becomes $\tau_{xx} = (2.5u_*)^2$, $\tau_{yy} = (2.3u_*)^2$, $\tau_{zz} = (1.3u_*)^2$, $\tau_{xz} = -(u_*)^2$, and $\tau_{xy} = \tau_{yz} = 0$. The mean dissipation rate of TKE is $\varepsilon = 5.7u_*^3/\kappa z$.

To obtain near statistically stationary concentration estimates, 420,000 particles were continuously released from a point
source at an emission rate of 200 particles per second with a timestep of 1 s ($\Delta t = 1$ s) for a duration of 2100 s. The power-law exponent for the velocity profile was 0.15 with a reference velocity $U = 5.90$ m s$^{-1}$ at a reference height of $H = 4$ m. The concentration was averaged over 1800 s with a starting time of 300 s after the beginning of the release. The number of sampling boxes in the $x$-, $y$-, and $z$-directions were 36, 59, and 20, respectively, over a domain size of 200 m $\times$ 100 m $\times$ 20 m. The source was specified to be at $x_s = 20$ m, $y_s = 50$ m and $z_s = H = 4$ m. The simulation parameters are summarized in Table 3.
Figure 6 shows the lateral and vertical normalized concentration profiles at four streamwise locations ($x/H = 4.03$, $x/H = 10.97$, $x/H = 19.31$, and $x/H = 37.36$) for the QES-Plume model and the non-Gaussian analytical solution. The model concentrations are in good agreement with the analytical solution. At all locations, the horizontal spread of the concentration along the centerline ($z/H = 1$) is captured well by QES-Plume. At $x/H = 4.03$, the concentration from QES-plume matches the analytical solution except at the peak of the plume where there is a $19.69\%$ error on the maximum. At the second profile
downstream of the release ($x/H = 10.97$), the QES-Plume concentration peak has a significantly improved match with the theoretical model with only a $2.6\%$ overprediction. Further downstream, the prediction below the centerline ($z/H = 1$) exhibits significant deviations from Eq. (28) and at $x/H = 19.31$ and beyond the model fails to reproduce both the height and the value

**Table 3.** Simulation parameters for continuous release in power-law boundary layer flow test case

| | |
|---|---|
| Domain size $(L_x, L_y, L_z)$ | 200 m, 100 m, 20 m |
| Reference velocity $(U)$ | 5.9 m s$^{-1}$ |
| Reference Height $(H)$ | 4 m |
| Power law exponent $(n)$ | 0.15 |
| Stress tensor correction $(C_u, C_v, C_w)$ | 2.5, 2.3, 1.3 |
| Advection timestep $(\Delta t)$ | 1 s |
| Concentration averaging time $(T)$ | 1800 s |
| Source location $(x_s, y_s, z_s = H)$ | 20 m, 50 m, 4 m |
| Source strength $(Q)$ | 200 particles/s |
| Sampling boxes $(\Delta x_s, \Delta y_s, \Delta z_s)$ | 5.5 m, 2 m, 1 m |

of the peak. The QES-Plume predictions exceed the theoretical concentration at the ground by a factor 3 at $x/H = 19.31$. Additionally, the profile calculated from the model shifts to a monotonous profile around $x/H = 23.47$ whereas the analytical solution does not exhibit the same behaviour (Fig. 6 h). The non-Gaussian solution is known to underpredict the concentration close to the ground (Brown et al., 1993, 1997) likely accounting for a significant amount of the observed deviations in this region.

### 4.3 Array of cubical buildings

The idealized test cases are useful to evaluate QES-Plume's GLEs solution methodology because they control for external factors that impact the quality of the dispersion model including turbulence parameterization, mean velocity specification, and boundary conditions. The limitation is that they do not fully engage the GLEs because they do not activate all components of the stress and velocity gradient tensors. To more fully examine the performance of QES-Plume's GLE implementation, it is compared to dispersion data from a wind-tunnel experiment for a $7 \times 11$ array of cubical buildings. The wind-tunnel experiment was conducted in a United State Environmental Protection Agency (EPA) meteorological wind tunnel (Brown et al., 2000; Lawson et al., 2000; Brown et al., 2001). For concentration estimates, high-purity ethane ($C_2H_6$, chemically pure grade, minimum purity 99.5 mole percent) was used as a tracer, which is slightly heavier than air (molecular weight 30). This tracer may be regarded as neutrally buoyant owing to the high turbulence level and the release rate of the tracer. A perforated plastic sphere with a diameter of 10 mm was used for continuously releasing the tracer at ground level behind the first centerline building of the $7 \times 11$ array. Figure 7a presents the layout of the array within the domain. Figure 7b shows the location of the source (diamond) and the location of vertical concentration profile measurements (circle) in the wind-tunnel. In addition, spanwise transverse concentration measurements were made at a height of $z/H = 0.1$ at the same $x/H$ locations, except at $x/H = 2.5$ because this location interferes with the buildings. The minimum quantifiable non-dimensional concentration for the wind-tunnel measurements is assumed to be $10^{-3}$ (following the recommendation from Chang and Hanna, 2004).

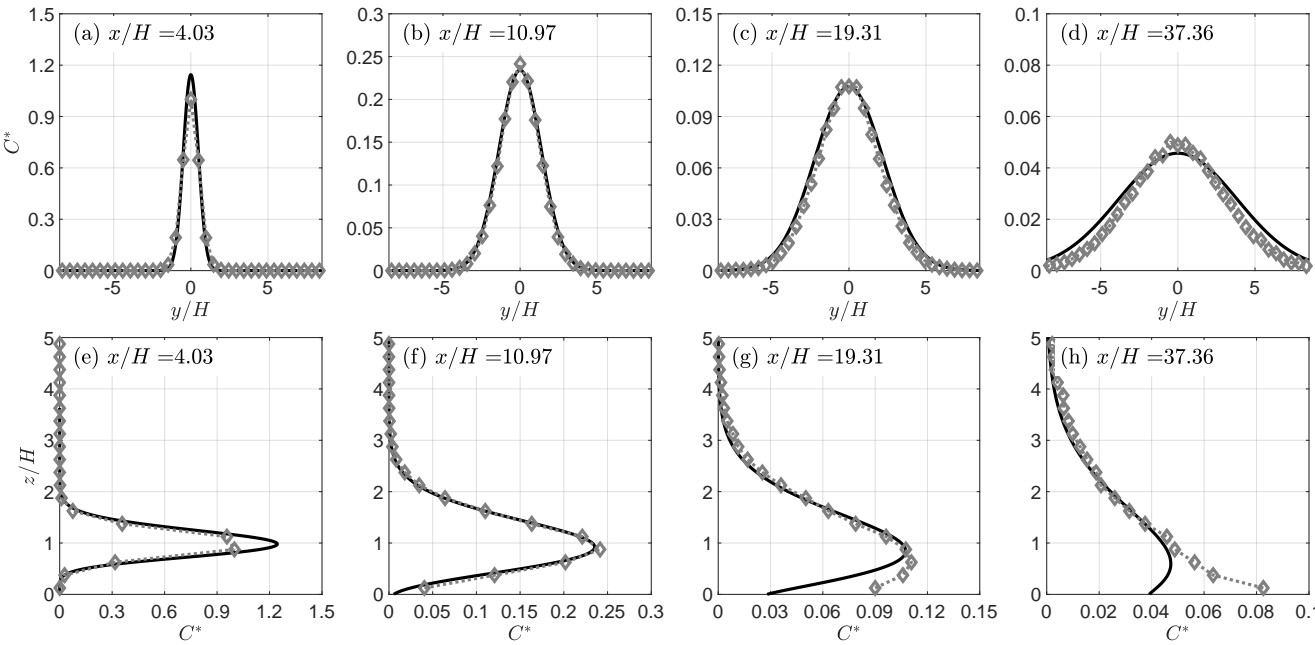

**Figure 6.** Profiles of the normalized concentration for the non-Gaussian plume analytical solution (line) and the QES-Plume model (diamond) at four different $x/H$ locations. The top panels are lateral profiles at $z/H = 0.875$, and bottom panels are vertical profiles at $y/H = 0$. Concentrations are normalized following Eq. (19).

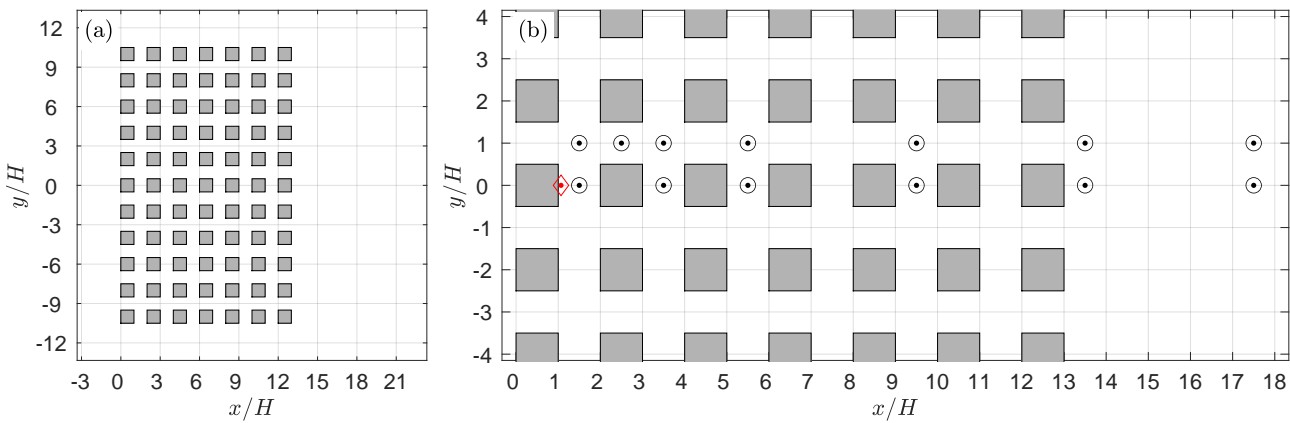

**Figure 7.** Simulation setup for the $7 \times 11$ array of cubical buildings. Panels are: (a) full simulation domain of $26.6H \times 26.6H$, and (b) zoomed-in section of the domain showing the location of the source behind the first building at $z/H = 0.1$ (diamond) and the locations of wind-tunnel measurements at $y/H = 0$ and $y/H = 1$ for multiple downstream locations (circle).

**Table 4.** Simulation parameters for the array of cubical buildings test case

| | |
|---|---|
| Domain size $(L_x, L_y, L_z)$ | 400 m, 400 m, 60 m |
| Inflow velocity $(U)$ | 2.83 m s$^{-1}$ |
| Reference height $(H)$ | 15 m |
| Aerodynamic surface roughness $(z_0)$ | 0.02 m |
| Stress tensor correction $(C_u, C_v, C_w)$ | 2.5, 2.0, 1.3 |
| LSDM constant $(C_0)$ | 5.7 |
| Non-local mixing coefficient $(C_{nlm})$ | 0.3 |
| Advection timestep $(\Delta t)$ | 1 s |
| Concentration averaging time $(T)$ | 3600 s |
| Source strength $(Q)$ | 500 particles/s |
| Sampling boxes $(\Delta x_s, \Delta y_s, \Delta z_s)$ | 1.5 m, 1.5 m, 1.5 m |

For this test case, QES-Plume was driven using the flow field computed by QES-Winds. Following Singh et al. (2008),
the street-canyon parameterization needed to be modified to address some of the shortcomings of the original street-canyon parameterization from Röckle (1989) used by QUIC-URB. The modified model adds a blending region at the edge of the canyons to resolve issues related to erroneous gradients, a feature of the flow which is critical to obtain the correct particle dispersion in and out of the street canyons. The turbulence fields were computed by QES-Turb, with the addition of a non-local background mixing (Sec. 3.2).

A total of 1,950,000 particles were released continuously for 3900 s (i.e., $Q = 500$ particles per second) from a point source to obtain near statistically stationary concentration estimates with QES-Plume. The concentration was averaged over 3600 s with a starting time of 300 s after beginning the release. The size of the sampling boxes in $x$-, $y$- and $z$-directions was set to 1.5 m, 1.5 m and 1.5 m, respectively, over a domain size of 400 m × 400 m × 60 m. The source was placed behind the first centerline building of the array at $x/H = 1.067$, $y/H = 0$ and $z/H = 0.067$ to be in agreement with the wind-tunnel
experiments. During the simulation, no "rogue" trajectories were detected, confirming the stability of the integration scheme in a complex environment.

Cross sections of the concentration results of dispersion through the building array are shown in Fig. 8. Many processes characteristic of urban dispersion at the neighborhood scale are illustrated (Belcher, 2005). To simplify the discussion, the spaces in-between the buildings along the $x$- and $y$-directions are designated as street channels and street canyons, respec-
390 tively. Importantly, the minimal non-dimensional concentration computed by the model is $C_* = 1.5 \times 10^{-4}$. For comparison with the wind-tunnel data, values smaller than the experimental minimum threshold of $10^{-3}$ –marked by the gray contour line in the figure– are considered as zeros. Figure 8a shows a vertical slice along the $y/H = 0$ centerline and Figure 8b a horizontal slice at $z/H = 0.3$. As particles advect downstream, the plume spreads laterally into the adjacent street canyons. Between the first and the third street canyons, the plume width grows linearly in the first street channel ($y/H = \pm 1.5$). This
linear growth stage is characteristic of the near-field region (Belcher, 2005). In addition, the recirculating flow present in the

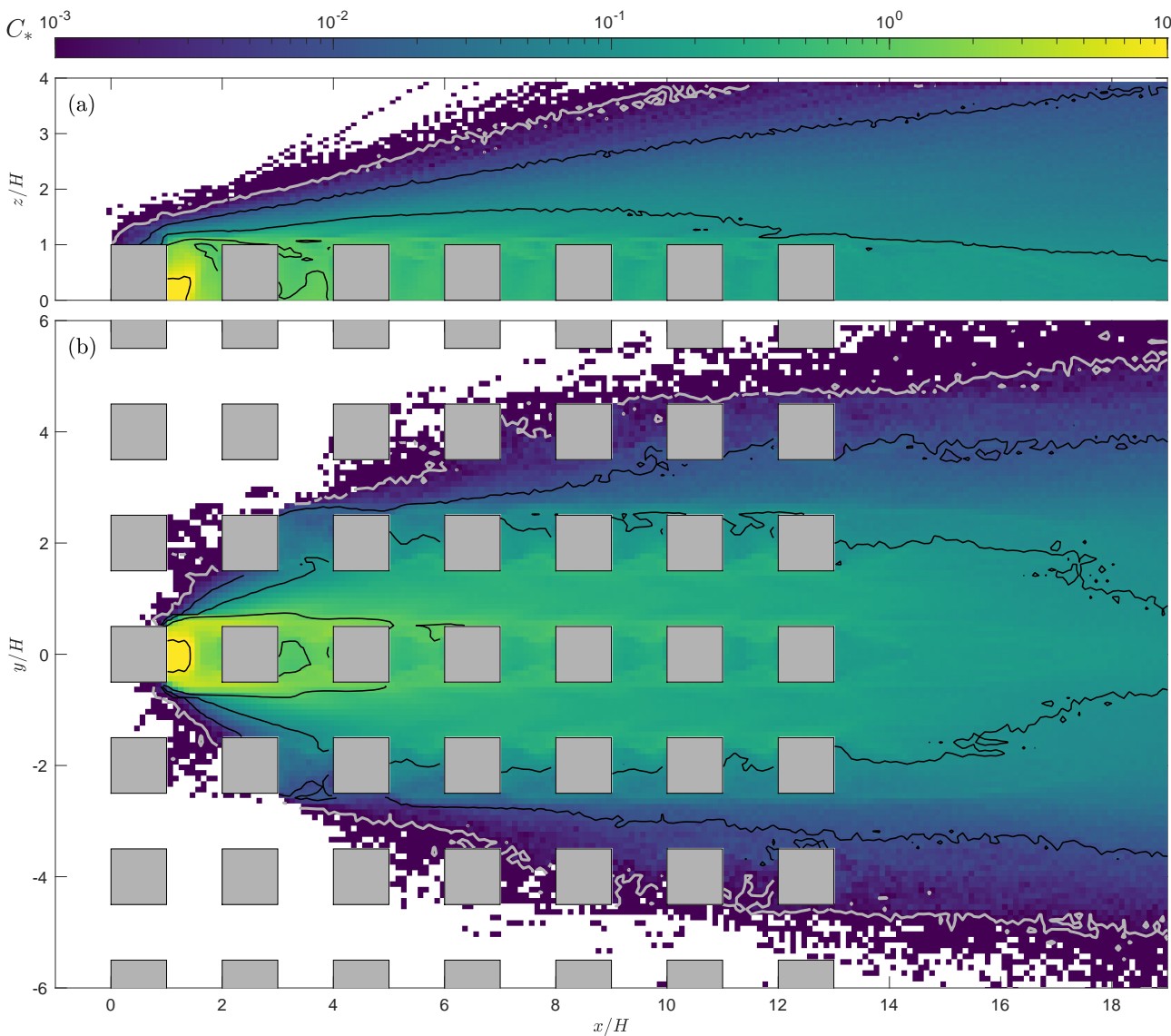

**Figure 8.** Non-dimensional concentration field from QES-Plume: (a) vertical slice at $y/H = 0$ and (b) horizontal slice at $z/H = 0.3$. The concentration has been normalized using Eq. (19), the color axis is in logarithmic scale, the contour lines represent each decade of the concentration scale, and the gray contour line ($C_* = 10^{-3}$) corresponds to the minimum concentration threshold measurement in the wind-tunnel experiment.

street canyons traps particles within the regions between buildings. High concentrations in street canyons behave similar to a source, leading to a phenomenon referred to as a secondary source (Belcher, 2005). Above the array, the vertical spread of the plume is enhanced by the vertical transport out of each street canyon. Farther downstream, the plume width is nearly constant

once the particles have reached the second street channel ($y/H = \pm 2.5$). Belcher (2005) calls this region the far field, it is characterized by a slower rate of spread past the first few rows of buildings. Street intersections also play an important role in the observed dispersion pattern. For example, T-junctions lead to very different behavior than four-way intersections. In the former, topological dispersion occurs due to dividing streamlines, which is not observed in the latter type of intersections (Belcher, 2005). In the array considered in the current simulation, there is no topological dispersion at the street intersections. Additionally, the near-field and far-field regions seem akin to the different scaling region presented in Fig. 5, suggesting that the region between $1 < x/H < 6$ and $-1.5 < y/H < 1.5$ corresponds to region of linear growth with $\tau_L < 1$. However, further investigation is needed and would be beyond the scope of this paper.

To begin a quantitative comparison, QES-Plume results are plotted with respect to the concentrations from the wind-tunnel dataset in Fig. 9 for vertical measurements along the centerline ($y/H = 0$, top panels) and in-between the row of buildings ($y/H = 1$, middle panels) as well as transverse measurements close to the ground in the street canyons ($z/H = 0.1$, bottom panels) for multiple downstream locations. In the first street canyon (first panel of Fig. 9a), the concentration is measured in the middle of the street canyon, close to the source. The experimental profile increases from the ground to its maximum near the building height ($z/H = 0.85$) and then decreases with height. In contrast, the QES-Plume profile contains two maxima, one in the middle of the street canyon ($z/H = 0.40$) and one above the top of the building ($z/H = 1.1$). The location of the street-canyon vortex and the shear-layer growth are critical to the dynamics of dispersion in the first street canyon because of the source location. Small deviations can have large consequence in the concentration profile. A stronger back flow in the lower part of the street canyon is observed in the wind tunnel, leading to the higher location of the maximum concentration compared to QES-Plume. The vertical spread above the building height is comparable between the QES-plume and the wind-tunnel data (see Appendix A1). Overall, the model prediction is acceptable considering the strong sensitivity to source location with an RMSE of the normalized concentration of 2.305 (relative RMSE of 0.272). The RSME and relative RMSE of all the different profiles are compiled in Table 5.

In the second street canyon ($x/H = 3.5$), the experimental profile is mostly constant between the ground and $z/H = 0.5$, then increases to reach its maximum at $z/H = 1.1$, and finally decreases to the top of the plume at $z/H = 2.0$. The profile calculated by QES-Plume shows a significantly different shape with two large spikes at $z/H = 0.8$ and $z/H = 1.1$. The QES concentration close to the ground is 50% higher than the experimental concentration and is not constant in the lower part of the street canyon. It decreases until $z/H = 0.7$ then spikes at $z/H = 0.8$. Interestingly, there is only a 2% difference between the mean concentrations within the street canyon. Hence, even if QES does not capture the shape of the profile, the overall average concentration is estimated correctly. The different spikes observed in the profile are linked to the parameterization used by QES-Winds, an over-estimation of the velocity variances, and the presence of sharp gradients at the top of the street canyon ($0.7 < z/H < 1.1$). A similar shape is observed for the concentration profile at the center of the third street canyon ($x/H = 5.5$) with an RMSE equal to 0.098. In addition, the average concentrations in the street canyon are also within a 2% margin. The shape of the profile in the fifth street canyon ($x/H = 9.5$) is very similar to the profile discussed previously but with the experimental values consistently higher than values calculated by QES-Plume. Although the values match close to the surface, the mean experimental concentration in the canyon is 20% higher. Likewise, QES-Plume significantly under-estimates

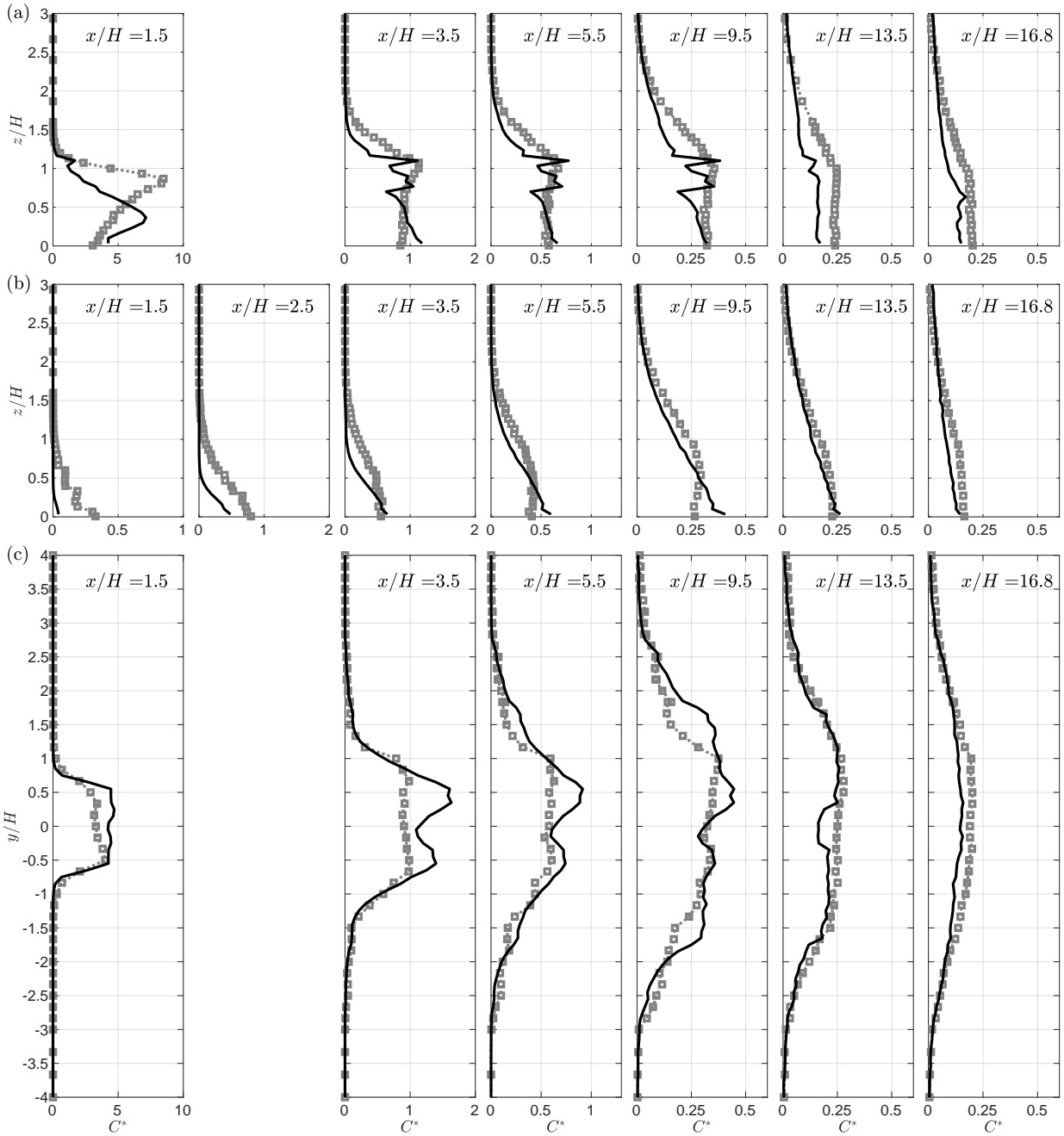

**Figure 9.** Comparison of QES-Plume concentration (lines) with wind-tunnel data (squares) for the $7 \times 11$ cubical array study. The panels in (a) show vertical profiles at $y/H = 0$, the panels in (b) show vertical profiles at $y/H = 1$, and the panels in (c) show horizontal profiles at $z/H = 0.1$. The downstream locations are reported in the panels. The concentrations have been normalized using Eq. (19). Missing panels correspond to locations where the measurements would have coincided with buildings.

the concentration above the buildings from $z/H = 1$ to $z/H = 2$. The rest of the profiles are in good agreement. At this location, the RMSE is equal to $0.068$. Past the last building ($x/H = 13.5$), the shape of the calculated profile does not exhibit large spikes and is closer to the experimental data. However, the model consistently under-estimates the concentration at this location, leading to an RMSE of $0.075$. In the far wake ($x/H = 16.8$), the model under-estimates the concentration compared to the experimental data with an RMSE of $0.048$. The oscillations observed at the last two locations correspond to the transition between the different wake regions used by QES-Winds. For example, the spike at $z/H \approx 0.6$ for $x/H = 16.8$ corresponds to the edge of the far-wake parameterization.

The middle panels (Fig. 9b) show the vertical profiles in the first street channel ($y/H = 1$). At $x/H = 1.5$, the model under-estimates the concentration, predicting almost zero concentration at all heights. An indication of the lack of lateral dispersion into the channel. At $x/H = 2.5$, the modeled concentration is no longer zero but still under-estimates the data, with concentrations about half of the wind-tunnel measurements for $z/H < 0.5$ and almost zero for $z/H > 0.5$, resulting in a RMSE $= 0.197$. Farther downstream, at $x/H = 3.5$, QES-Plume shows significant lateral dispersion and is in good agreement with the experimental data close to the ground ($z/H < 0.2$). However, the simulated concentrations decrease much faster with height compared to the wind-tunnel data for $z/H > 0.2$ and significantly under-estimate concentrations for $z/H > 0.5$ to $z/H < 1.5$, resulting in a RMSE value of $0.130$. At $x/H = 5.5$, the measured concentration profiles are below $z/H < 0.5$ and decrease to reach zero at $z/H \approx 1.5$. The simulated concentrations exhibit a similar behavior, even if the lower plateau is smaller ($z/H < 0.3$). They also reach zero at roughly the same height, resulting in a RMSE of $0.070$. At $x/H = 9.5$, QES-Plume predictions are in relatively good agreement with wind-tunnel measurement with RMSE $= 0.034$. However, the measured concentration profile is not monotonous, unlike the QES-Plume profile, with a maximum at $z/H \approx 0.75$. The model failed to capture this feature of the profile. Past the last building, at $x/H = 13.5$ and $x/H = 16.8$, the simulated concentrations show good agreement with the wind-tunnel measurements for all heights with RMSE $= 0.018$ and RMSE $= 0.028$, respectively.

Lateral measurements of the concentration are presented in Fig. 9c. In the first street canyon, the concentration from QES-Plume contains two spikes at $y/H = \pm 0.5$ which are not observed in the experimental data. These spikes are located at the edges of the upstream building. The QES-Winds canyon model does not extend past the edge of the building, (Singh et al., 2008) which leads to sharp gradients at these locations, preventing some lateral dispersion. The concentration at the center of the street canyon and the width of the plume match between the model and the experiment. The RMSE over the whole profile is $0.335$. Similar behavior is observed in the second and third street canyons with RMSE $= 0.152$ and RMSE $= 0.077$, respectively. Farther downstream, the fifth street canyon is past the near field and the plume has spread beyond the second street channel and into the third. At this point, the modeled and measured concentrations are in good agreement with a RMSE $= 0.034$. The oscillations observed in the simulated data are linked to the different parameterization used by QES-Winds. Past the last building, the plume has spread from $y/H < -3$ to $y/H > 3$. Although, QES-Plume still has some under-estimation of the concentration, the total width of the plume matches well.

Some of the discrepancies found in Fig. 9b are linked to the sharp concentration drop-off observed in the horizontal profiles which, for locations in the first half of the array where the plume width is growing linearly (near field), can be found at the middle of the first channel in between the buildings ($y/H = 1$). The location of this drop-off is very sensitive to velocity and

**Table 5.** Summary of QES-Plumes's RMSEs and relative RMSEs at various locations for the $7{\times}11$ cubical array study.

| Location | $x/H$ | 1.5 | 2.5 | 3.5 | 5.5 | 9.5 | 13.5 | 16.8 |
|---|---|---|---|---|---|---|---|---|
| $y/H = 0$ | | 2.305 | | 0.187 | 0.098 | 0.068 | 0.075 | 0.048 |
| $y/H = 1$ | | 0.087 | 0.197 | 0.130 | 0.070 | 0.034 | 0.018 | 0.028 |
| $z/H = 0.1$ | | 0.335 | | 0.152 | 0.077 | 0.049 | 0.023 | 0.023 |
| $y/H = 0$ | | 27.2% | | 16.5% | 14.6% | 19.1% | 30.2% | 23.7% |
| $y/H = 1$ | | 26.7% | 24.7% | 22.6% | 16.2% | 11.8% | 7.7% | 17.0% |
| $z/H = 0.1$ | | 8.4% | | 15.5% | 12.2% | 13.2% | 8.3% | 11.7% |

turbulence fields. In the second half of the array, a sharp drop-off is not observed in the horizontal concentration profiles.
Overall, QES-Plume predicts well the width of the plume both within the array and past the last building as well as the value of the concentration. The mean relative RMSE over all the transects and profiles is 15.6 %.

Figure 10 shows the signed relative error between the QES-plume and the wind-tunnel data, where positive values represent overestimation by the model. Large relative errors can be observed at the edge of the plume, where the absolute concentrations are close to the lower limit of the measurements. The deviations observed in the first street canyon ($1 \leq x/H \leq 2$ and $-0.5 \leq$
$y/H \leq 0.5$) are linked to the sensitivity of the mean flow model related to the location of the street canyon vortex, emphasizing the discussion related to the first panel of Fig. 9a. Both vertical and horizontal slices show that large relative errors are found in the shear zones behind buildings, in particular at $y/H = 1.5$ between $x/H = 3$ and $x/H \leq 10$. These observations are related to the parameterization used by the wind model, where different zones are defined based on geometrical considerations (Singh et al., 2008). A comprehensive comparison between the mean velocity field and velocity variances from QES to the wind-
tunnel data is presented in Appendix A. Moreover, Figs. 9 and 10 illustrate the inherent asymmetry present in the data and model due to the variability of the system, where small deviations between the left ($y/H > 0$) and right ($y/H < 0$) sides can yield significant relative errors between the data and model.

Finally, the local-mixing turbulence model relies heavily on the magnitude of the local velocity gradients (see Eqs. 10 and 6). This is problematic in regions where velocity gradients are small and the model predicts negligible turbulence. On the
485 other hand, regions with sharp velocity gradients lead to unrealistically large stresses and TKE, due to the use of a diagnostic wind model and a local-mixing model, as illustrated in Appendix A. A future improvement to the turbulence model would be to limit magnitude of the velocity gradients, especially the vertical gradient of streamwise velocity. In addition, the appendix presents the importance of adding non-local mixing to the turbulence model to enhance lateral and vertical spread. Although this observation is related to wind-tunnel data, similar considerations hold for atmospheric flows, where background turbulence
levels need to be evaluated from sources such as meteorological measurements or weather prediction models.

In summary, QES-Plume is capable of reproducing concentration levels in this complex mock-urban setting despite weak performance in the first street canyon. In particular, the results show the expected linear behavior near the source and constant region farther away as reported by Belcher (2005). Similarly, critical metrics such as plume width and height are reproduced by the model. Specifically, out of the 894 measurements in the experimental data, 156 are below the concentration threshold and

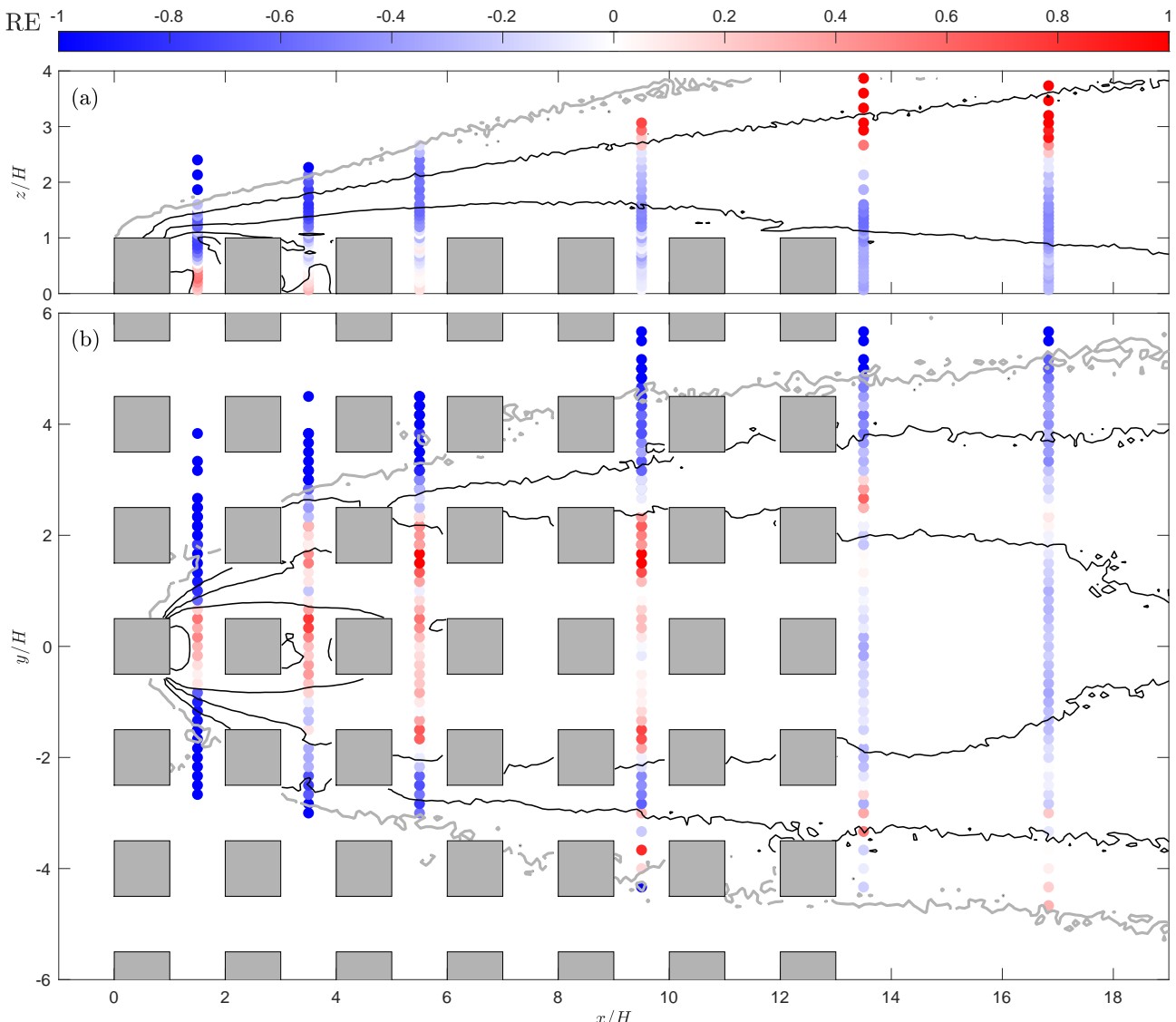

**Figure 10.** Signed relative error (RE) between the concentration field from QES-Plume and the wind-tunnel measurements where positive values (red markers) represent an overestimation by the model. The panels in the figure are (a) vertical slice at $y/H = 0$ and (b) horizontal slice at $z/H = 0.1$. The contour lines represent each decade of the concentration form 10 to $10^{-3}$, and the gray contour line ($C_* = 10^{-3}$) corresponds to the minimum concentration threshold measurement in the wind-tunnel experiment.

QES-Plume matches $154$ of those points, yielding $99\%$ matched-zeros. Figure 11 shows a paired scatter plot of the vertical and horizontal concentration profiles. The predicted concentrations by QES-Plume are represented on the abscissa and the wind-tunnel concentration data is represented on the ordinate of the scatter plot. This plot shows that the model under-estimates

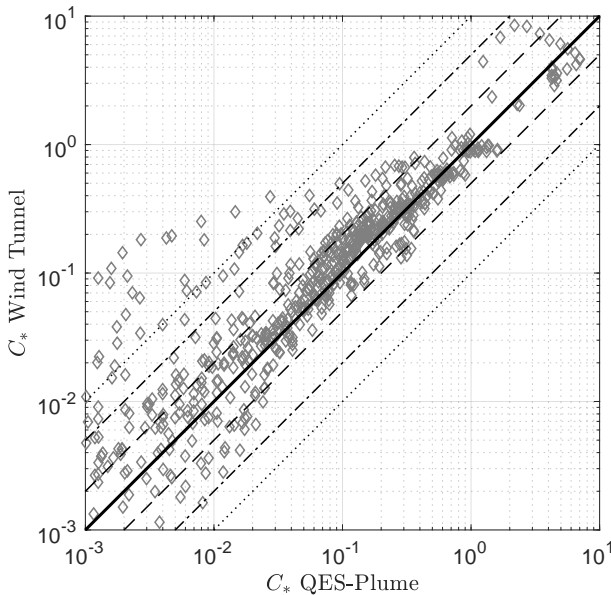

**Figure 11.** Paired scatter plot of the wind-tunnel concentration data and QES-Plume concentrations for the $7\times11$ cubical array study. The one-to-one line is represented by the black line, a factor of 2 by the dash lines, a factor of 5 by the dot-dashed lines, and a factor of 10 by the dotted lines.

**Table 6.** Summary statistics comparing QES-Plume results with the wind-tunnel data for the $7\times11$ cubical array study.

| Class | points in class/total | RMSE |
|---|---|---|
| Factor 2 | 433/740 (58.5%) | 0.300 |
| Factor 5 | 561/740 (75.8%) | 0.529 |
| Factor 10 | 594/740 (80.3%) | 0.519 |
| Match zeros | 154/156 (99%) | |
| False positive | 2/740 (0.3%) | |
| False negative | 117/740 (15.8%) | |

the concentration with 79% of QES concentration being lower than the experimental data. However, 59% of the predicted concentrations fall within a factor of 2 (with an RMSE of 0.300), 76% within a factor 5 (with an RMSE of 0.529), and 80% within a factor 10 (with an RMSE of 0.519). Of the 20% of data points outside of the factor 10 threshold, QES-plume failed to compute any concentration at 80% of these locations, corresponding to a 15% false negative rate. Statistics are summarized in Table 6. In addition, the RMSEs in the factor 5 and factor 10 are dominated by 5 points in the profile at $y/H = 0$ and $x/H = 1.5$. If the corresponding profile is removed from the calculation of the RMSEs, the values becomes 2 to 5 times smaller. All the data points outside of the factor 10 class correspond to normalized concentrations smaller than $5 \times 10^{-2}$, and coincide to the edge of the plume where both measurement and simulated values have large uncertainties.

Finally, QES-Plume exhibits excellent computational performance even without parallelization. The $7 \times 11$ array simulation runs for 3900 s of simulation time and contains about 88000 active particles at each timestep. The average total execution time on an AMD EPYC 7543 32-core processor is around 1100 s. This is equal to an execution time of less than $4 \times 10^{-6}$ s per particle per timestep.

## 5  Discussion

The method implemented to partially solve the stiffness problem from the GLEs (Sec. 3.3) guarantees numerical stability of the velocity fluctuations, which means that no energy is artificially added to the system. However, large gradients in the velocity field or regions of unrealistically large turbulence quantities can lead to nonphysical fluctuations. Equation (15) contains both temporal and spatial derivatives of the stress tensor, and therefore, the model remains sensitive to large increments in these two quantities. The same observation can be made for the mean dissipation rate of TKE. This quantity needs to stay within realistic bounds, especially since it multiplies the random term in the model. Finally, the particle positions are updated using Euler's forward scheme, which does not guarantee numerical stability. The authors have not encountered stability issues related to the last step of the model if the fluctuations obtained from the GLE take realistic values. The numerical stability issues associated with Eq. (17) do not lead to a degradation of the solution as truncation errors are not propagated in time.

The test cases presented by Bailey (2017) showed that the model produced good results even for large timesteps. However, some aspects of the model are sensitive to timestep sizes. Concentration calculations are impacted if the size of the collection box is too small for the particle motion. Similarly, the reflective boundary condition can lead to errors. For example, if the particle displacements exceed the size of some obstacles in the domain, the particles might appear to have warped through the obstacle.

The local-mixing turbulence model relies heavily on the magnitude of the local velocity gradients (see Eqs. 10 and 6). This is problematic in regions where velocity gradients are small and the model predicts negligible turbulence. On the other hand, regions with sharp velocity gradients lead to unrealistically large stresses and TKE. Appendix A presents the mean velocity field and velocity variances for the array of cubical buildings. Comparison with the wind-tunnel data is also included in the appendix. The results illustrate the consequences of using a local-mixing model with a diagnostic wind solver. A future option would be to limit magnitude of the velocity gradients, especially the vertical gradient of streamwise velocity. In addition, the appendix presents the importance of adding non-local mixing to the turbulence model to enhance lateral and vertical spread. Although this observation is related to wind-tunnel data, similar considerations hold for atmospheric flows, where background turbulence levels need to be evaluated from sources such as meteorological measurements or weather prediction models.

## 6  Conclusions

Due to the presence of a large number of terms in the GLEs, SLEs are employed in most mainstream dispersion models. The presence of numerical instabilities due to the stiffness of the GLEs has also been a problem for the explicit integration

of the GLEs into Lagrangian dispersion models. Although commonly used numerical methods for solving the SLEs are still numerically unstable, the SLEs are considered slightly more stable compared to the GLEs due to the drastic reduction in the number of terms. This paper discussed the implementation of the GLEs with the implicit time-integration method from Bailey (2017) to alleviate the stiffness problem from the GLEs and eliminate "rogue" trajectories (Yee and Wilson, 2007). This implicit scheme was implemented in the QES framework and is dynamically coupled with a mass-consistent wind model and a local-mixing turbulence model.

QES-Plume has been validated against analytical solutions in idealized conditions where the model yielded good results. In particular, the overall maximum relative error was under $6\%$ and the model captured horizontal and vertical plume width accurately. Discrepancies between the model and the analytical solutions matched known shortcomings of the latter. During the comparison with experimental data, QES-Plume performed well, highlighted by the concentration contours showing good levels of lateral and vertical dispersion, as well as a well-mixed plume in the street channels. The results showed $99\%$ matched-zeros, a factor 2 concentration prediction of $59\%$, and only a $15\%$ false negative rate. However, the $7 \times 11$ test case emphasized the sensitivity to the mean-wind and turbulence models, as emphasized by Bahlali et al. (2019). In particular, the street-canyon model proposed by Singh et al. (2008) is used in this study and yielded better mean flow, turbulence, and dispersion results compared to the base model proposed by Röckle (1989). Moreover, the local mixing-length turbulence model necessitated the addition of a constant to the diagonal elements of the stress tensor to enhance the turbulence mixing within the street canyons and in the free stream. The constant of $0.3 \text{ m}^2 \text{ s}^{-2}$ was added to match the turbulence level of the experimental data within the free stream. However, background turbulence levels should be adjusted to reflect realistic conditions and can be evaluated from meteorological measurements or weather prediction models. In general, the results are dependent on the wind and turbulence models. Further improvement of QES-Turb are planned to address some of the shortcomings observed in the turbulence fields, such as sharp gradients in the shear zone. In addition, QES-Plume does not require QES-Winds and QES-Turb, the code can be used with inputs from other models.

Finally, QES-Plume was implemented from the ground-up as an object-oriented C++ code and has demonstrated excellent computational performance. Future versions of QES-Plume are likely to use a GPU-based implementation to enable much faster than real-time simulations. The potential use cases of a model like QES-Plume are numerous. For example, the model can be used to run simulations for decision makers for tabletop exercises or to study particulate dispersion in complex environments, such as spore or smoke transport in agricultural fields, as well as urban pollution and air quality.

*Code availability.* The Quick Environmental Simulation (QES) software is developed as a collaboration between the University of Utah, University of Minnesota Duluth, and Pukyong National University (Margairaz et al., 2022a). The software is written in C++ and NVIDIA's CUDA. The latest version of QES is publicly accessible on GitHub (https://github.com/UtahEFD/QES-Public) under the GNU General Public License (version 3). More information is available on GitHub about the installation of QES with or without CUDA. QES version v2.0.0 was used to produce the results used in this paper and is archived on Zenodo (Margairaz et al., 2022a).

*Data availability.* The data from the EPA for the $7 \times 11$ array wind-tunnel experiment from Brown et al. (2000) is publicly available. The dataset can be obtained upon request to the EPA (https://cfpub.epa.gov/si/si_public_record_report.cfm?dirEntryId=63740&Lab=NERL). The dataset can also be downloaded from the QES-Public GitHub repository (https://github.com/UtahEFD/QES-Public/blob/main/paperData/BROWN2000.zip).

## Appendix A: Modeled mean wind and turbulence fields for the $7 \times 11$ array of cubical buildings

In this section, the performance of QES-Winds and QES-Turb is briefly discussed in the context of the $7 \times 11$ array of cubical buildings. As dispersion results rely heavily on mean flow and turbulence stress fields, it is important to understand the quality of the fields used to drive the dispersion model. This section is not intended as a strict validation of the mean flow model (see Singh et al., 2008 for a validation of the street canyon model) or the local-mixing turbulence model.

### A1 Mean flow field

Figure A1 shows three profiles of the streamwise and vertical velocity upwind of the first building. The profiles at $x/H = -1$ illustrate that the inlet velocity field used in QES matches the experimental data. An upstream recirculation zone starts at $x/H \approx -0.8$ and grows to reach $z/H \approx 0.7$ on the face of the building. The QES profiles in this upwind region show the deceleration of the streamwise velocity and the displacement of the streamlines, in good agreement with the experimental data, on average. The $u$-velocity profile contains oscillations at the edge of the parameterization region, which is typical for the kind of model used. The vertical velocity at $x/H = -0.3$ illustrates the sensitivity to parameterization choices of the location of topological flow features such as the upwind vortex (Hayati et al., 2019). The experimental data indicate that this vortex is located further away from the building compared to the location predicted by QES. Hence, the $w$-velocity is positive in the lower part of the profile ($0 < z/H < 0.5$).

Figure A2 compares the profiles on top of the first building along the centreline of the array. Profiles at $x/H = 0.1$ illustrate that the flow field predicted by QES at this location is in good agreement with the experimental data. At $x/H = 0.5$, the streamwise velocity overshoots the experimental data at $z/H \approx 1.25$ as the rotation strength predicted by the rooftop vortex parameterization is higher than observed in the wind-tunnel data. Similarly, the parameterization over-predicts the $w$-velocity close to the back edge of the building $x/H = 0.9$.

Profiles at three locations in the first street canyon are presented in Fig. A3. The $u$-velocity profiles from QES contain sharp vertical gradients and a very shallow shear layer at the top of the street canyon ($z/H = 1$). In contrast, the experimental data has less steep gradients and a deeper shear layer. The street canyon model proposed by Singh et al. (2008) accounts for shear-layer growth from the edge of the building. However, in its current formulation, the model only parameterizes the part of the shear layer below $z/H = 1$ and does not extend vertically past the building top. Still, the street canyon model does a relatively good job of predicting the value of the streamwise velocity within the canyon. The vertical-velocity profiles illustrate that QES under-estimates $w$ in the second half of the street canyon. These observations indicate that the intensity of the street canyon vortex might be under-estimated and that its location might not be predicted correctly.

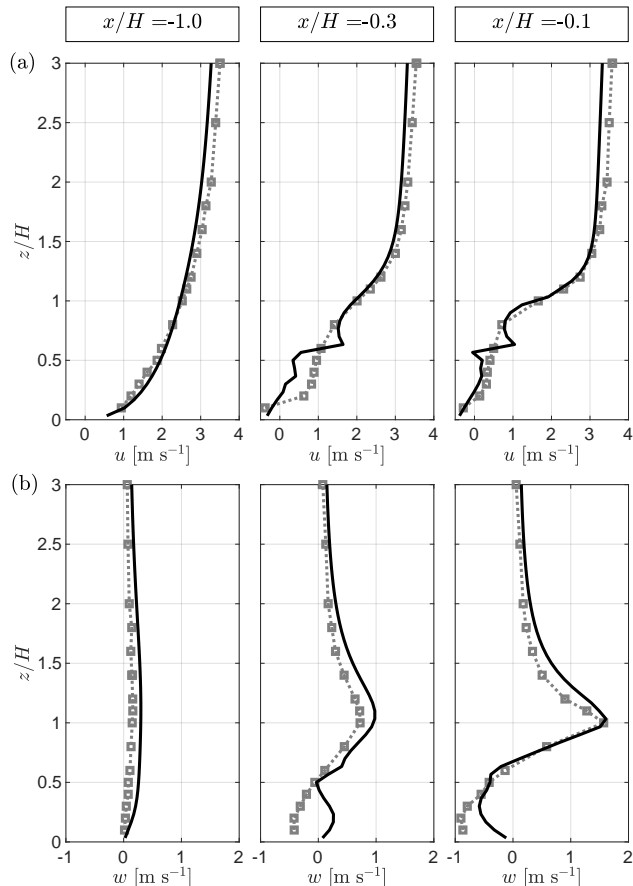

**Figure A1.** Comparison of QES-Winds (lines) with wind-tunnel data (squares) for the streamwise velocity $u$ (a) and the vertical velocity $w$ (b) upwind of the first building. All profiles are taken along the centerline ($y/H = 0$).

Figure A4 shows profiles downstream of the last building. Similar to the street canyon, the $u$-velocity profiles ($x/H = 13.1$ and $x/H = 13.5$) illustrate sharp vertical gradients close to the top of the building with a very shallow shear layer. The gradients are less steep in the experimental data and the shear layer is deeper, extending from $z/H \approx 0.8$ to $z/H \approx 1.5$. These observations are consequences of the formulation of the wake model as it does not extend vertically past the top of the building and does not model a shear layer that grows above the top of the building. At $x/H = 16.5$, the downstream recovery calculated by QES is faster than observed in the experimental data. Finally, the $w$-velocity profiles show good agreement for all profiles.

Overall, the mean flow along the centerline of the array is in good agreement with the experimental data and illustrates that the model proposed by Singh et al. (2008) has been correctly implemented in QES.

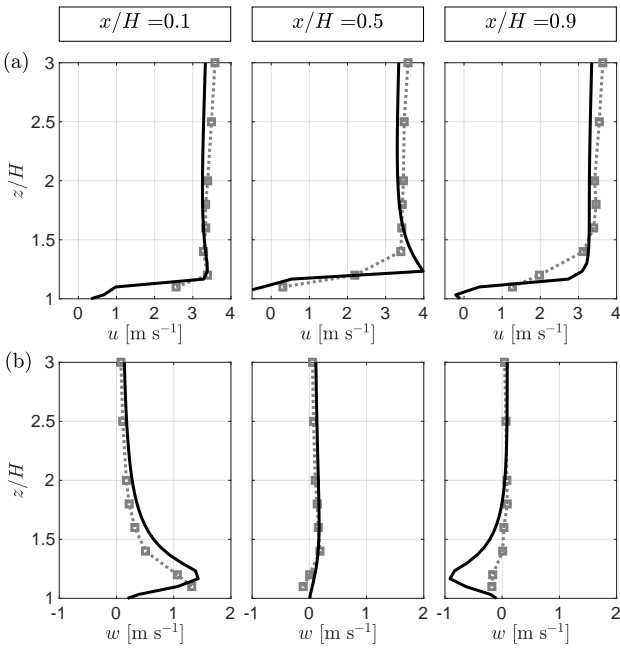

**Figure A2.** Comparison of QES-Winds (lines) with wind-tunnel data (squares) for the streamwise velocity $u$ (a) and the vertical velocity $w$ (b) at the roof-top level of the first building. All profiles are taken along the centerline ($y/H = 0$).

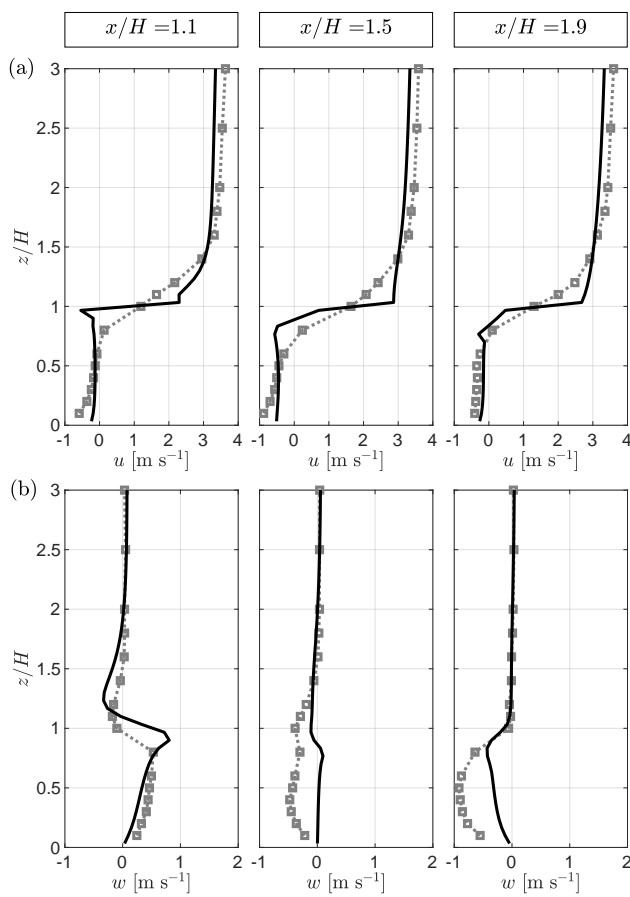

**Figure A3.** Comparison of QES-Winds (lines) with wind-tunnel data (squares) for the streamwise velocity $u$ (a) and the vertical velocity $w$ (b) within the first street canyon. All profiles are taken along the centerline ($y/H = 0$).

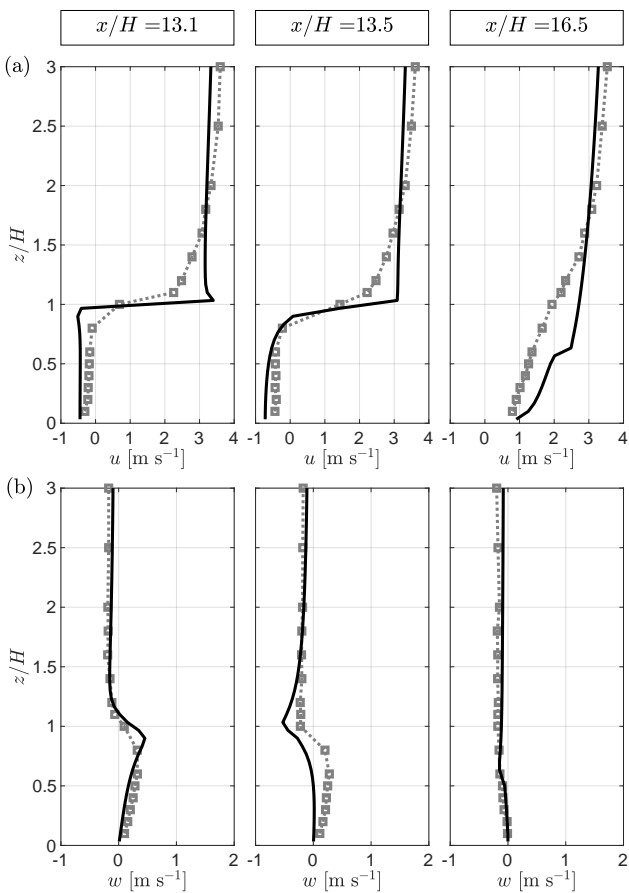

**Figure A4.** Comparison of QES-Winds (lines) with wind-tunnel data (squares) for the streamwise-velocity $u$ (a) and the vertical-velocity $w$ (b) downwind of the last building. All profiles are taken along the centerline ($y/H = 0$).

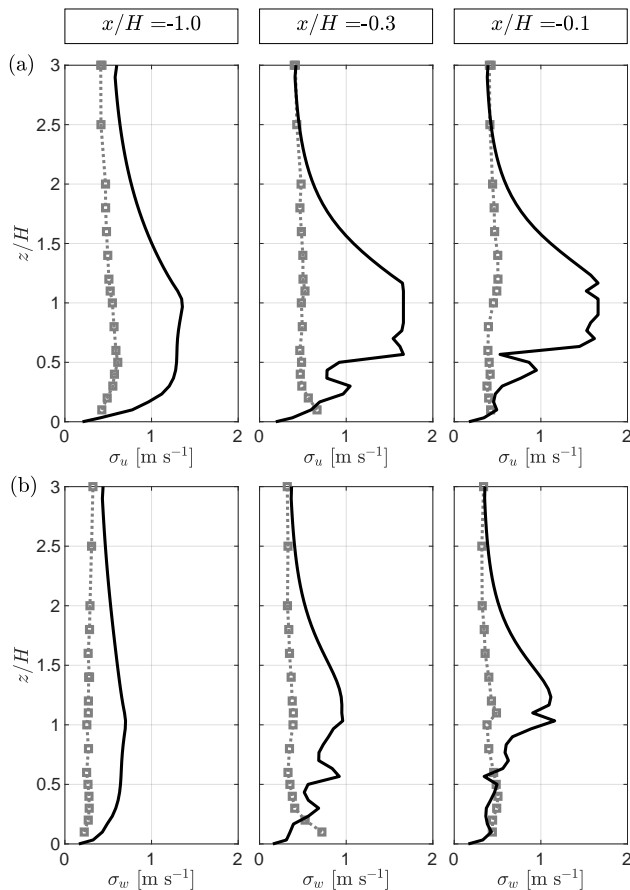

**Figure A5.** Comparison of QES-Turb (lines) with wind-tunnel data (squares) for the streamwise-velocity variance $\sigma_u$ (a) and the vertical-velocity variance $\sigma_w$ (b) upwind of the first building. All profiles are taken along the centerline ($y/H = 0$).

## A2    Turbulence fields

In this section, the velocity variances from the QES-Turb model and the wind-tunnel data are compared at selected locations to qualitatively evaluate the performance of the model.

Figure A5 indicates that QES-Turb over-estimates both the streamwise-velocity and the vertical-velocity variances upwind of the first and second buildings. Within the upwind recirculation zone ($-0.8 < x/H \leq 0$ and $0 \leq z/H < 0.7$), the model performs better. In addition, the signature of the oscillation observed in the velocity field can be seen in the variances as expected from Eq. (6).

Figure A6 compares the QES-Turb model and the wind-tunnel data for the velocity variances at the centerline over the rooftop of the first and second buildings. The profiles illustrate that QES overestimates the streamwise-velocity variance above the first building ($1 < z/H < 2$). The vertical-velocity variance is overestimated near the leading edge of the first building. The

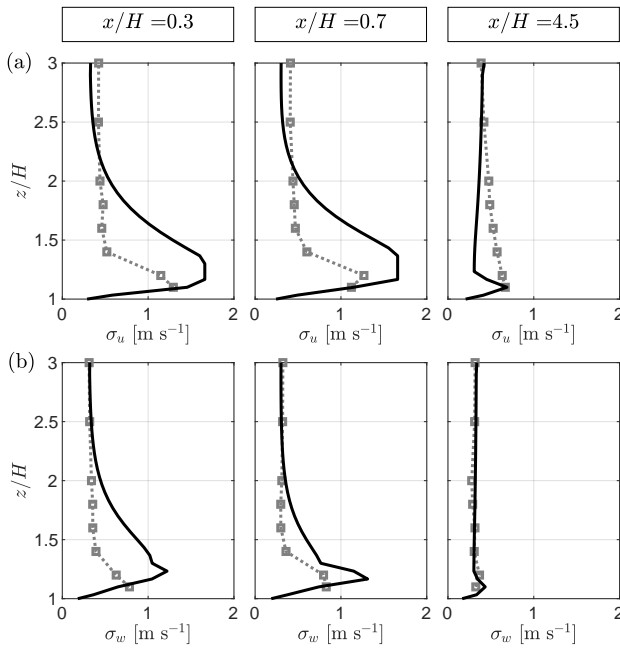

**Figure A6.** Comparison of QES-Turb (lines) with wind-tunnel data (squares) for the streamwise-velocity variance $\sigma_u$ (a) and the vertical-velocity variance $\sigma_w$ (b) at the roof-top of the first and second buildings. All profiles are taken along the centerline ($y/H = 0$).

model yields better results past the middle of the building. Over the second building, both variances are better reproduced by
620 the model. The same observation can be made at all available downstream roof-top locations (not shown). All profiles show
that the modeled free-stream turbulence levels match the experimental data well.

Figures A7 and A8 compare the QES-Turb model and the wind-tunnel measurements for the velocity variances at the
centerline of the first street canyon and downwind of the last building. The calculated variances are in good agreement with the
experimental data within the free stream for all profiles. The same observation can be made in the lower half of the street canyon
($z/H < 0.5$). However, the region in-between shows that the turbulence model over-estimates the variances. In particular at
$z/H = 1$, where the transition between the parameterizations occurs, the variances are over-estimated due to the steep gradients
presented in Sec. A1. Similarly, in the wake and recovery zones downstream of the last building, both variances contain a large
peak at this transition.

In summary, the turbulence model used to drive QES-Plume shows good agreement with the wind-tunnel data. Importantly,
the local-mixing model needed the addition of a constant $C_{nlm}$ to enhance the turbulence mixing within the street canyons
and in the free stream (see Sec. 3.2). Without the non-local mixing constant, particles had the tendency to follow closely the
mean wind near the source until they are ejected out of the first street canyon without much mixing within the street canyon
and channel. The constant was selected to match the turbulence level of the experimental data within the street canyons and
in the free stream. Although the turbulence levels at the edges of the street canyons are increased with the non-local mixing

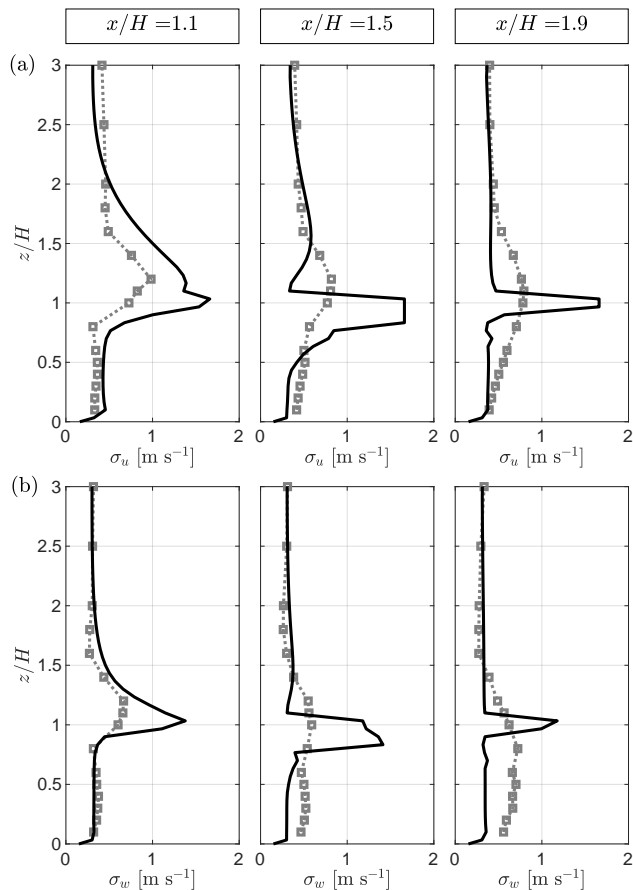

**Figure A7.** Comparison of QES-Turb (lines) with wind-tunnel data (squares) for the streamwise-velocity variance $\sigma_u$ (a) and the vertical-velocity variance $\sigma_w$ (b) within the first street canyon. All profiles are taken along the centerline ($y/H = 0$).

constant added, the turbulence levels within the street canyons matched the test data well. The result of this modification was to augment the magnitude of the turbulence within the street canyons while maintaining the influence of the velocity gradients in the stress tensor.

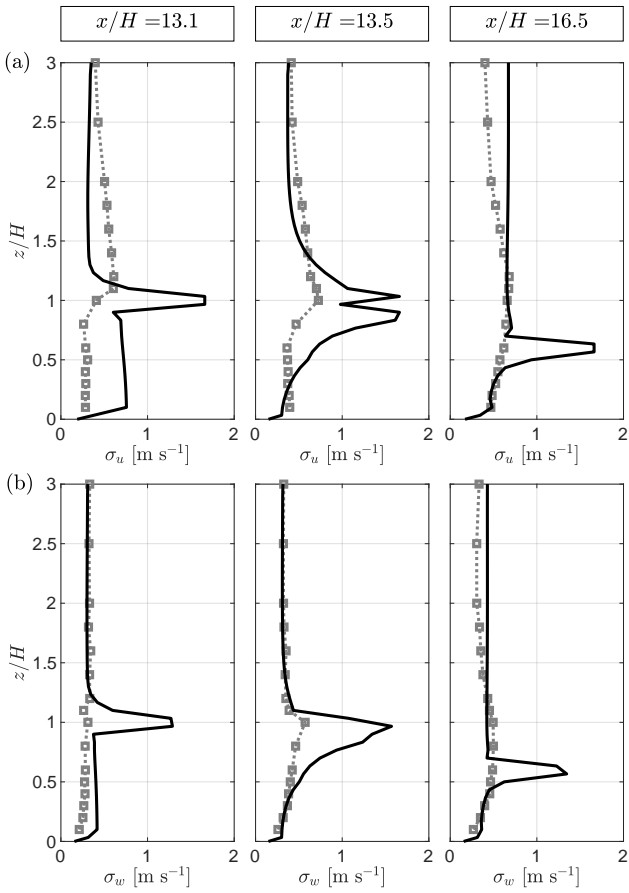

**Figure A8.** Comparison of QES-Turb (lines) with wind-tunnel data (squares) for the streamwise-velocity variance $\sigma_u$ (a) and the vertical-velocity variance $\sigma_w$ (b) downwind of the last building. All profiles are taken along the centerline ($y/H = 0$).

*Author contributions.* Conceptualization was done by RS and ERP. The code was written by JG, LA, and FM. Test cases were developed by BS and FM. The analysis and interpretation of the data were carried out by FM. The figures were produced by FM. The original draft of the paper was written by FM, with edits, suggestions, and revisions provided by JG, BS, RS, and ERP. Grants supporting the project leading to this publication were awarded to RS and ERP.

*Competing interests.* The authors declare that they have no conflict of interest.

*Acknowledgements.* The authors would like to acknowledge the funding provided National Institute of Environment Research (NIER), funded by the Ministry of Environment (MOE) of the Republic of Korea (NIER-SP2019-312), the United States Department of Agriculture National Institute for Food and Agriculture Specialty Crop Research Initiative (Award No. 2018–03375), and the United States Department of Agriculture Agricultural Research Service through Research Support Agreement (No. 58-2072-0-036).

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
