# Peer review of "QES-Plume v1.0: A Lagrangian dispersion model"

_EGUsphere, 2022_

## Author Comment (AC1)

**RC1**: 'Comment on egusphere-2022-1256', Bertrand Carissimo, 31 Jan 2023

*General comments*
The paper describes the improvement and validation of an operational model set that allow fast Lagrangian computation of pollutant dispersion in urban area. The paper is clear well-structured and well written. Only a few clarifications are needed.

The model and the validations are openly available as indicated.

My main regret is that the validation with analytical solutions is performed only with Eulerian solutions. For me it is therefore incomplete and this needs to be done in further work

Thanks for the comment. Regarding the validation against a Lagrangian solution, if the reviewer has a specific solution in mind that would improve confidence in QES-Plume's results, we would be happy to explore it. To address the review's concern, we have added a new analysis in section 4.1 (figure 5), where the trajectory's statistics are compared to the classical dispersion from a point source in statistically stationary isotropic turbulence from Pope and Taylor. The results show that QES reproduces both linear and square-root scaling for short-distance and long-distance transport.

*Specific comments*
l33 examples of Eulerian models given are large scale. Examples of micro scale obstacle resolving Eulerian models should be given.
We have added examples such as PALM and Code Saturne. We have adjusted the text accordingly.

l55: the unstable modes are in the numerical solution not the equations. Also, they are not really "modes" (idem l 108)
Thank you for the comment. We have corrected the manuscript to reflect this comment and changed to "numerical instabilities" and "methods numerically unstable" rather than "modes".

l 111: the top boundary conditions is mentioned here but not really discussed in the rest of the paper
We have added a comment related to the top boundary condition. The most common BC used here is an outlet condition, where particles are simply removed. However, in some cases, more complex top BCs need to be

considered. Our manuscript could have been misleading, and we changed the phrasing. We have also added a comment in section 3.3.1 about boundary conditions.

Table 1: anisotropy is taken into account for different terrain type (rural, urban, forest...) but there is no discussion of the impact of atmospheric stability. Why? Please give some argument and some discussion.

Thank you for the comment. The table's purpose is to provide a range of values for the anisotropy coefficient. This paper is focused on the urban canopy sublayer where the atmospheric stability plays a minimal role because the flow in that region is mostly neutral (for example see Ramamurthy et al. 2007). However, correction of these coefficients would be required in the surface layer, following Monin-Obukhov similarity theory. We have modified the text to address this comment.

L 185: the non-local background mixing coefficient is based on wind tunnel. How can we justify it for low wind speed strongly stratified real cases?

We cannot and the model should not be used in the cases mentioned by the reviewer. The non-local background mixing is not intended to be a universal model for all types of flows, with different atmospheric stability. Background mixing is a model input that needs to be specified to match different test cases and can be obtained from meteorological data such as turbulence measurements, weather models, or adjusted based on stability classes. We have added a comment to the text to reflect that the background mixing constant used is not intended to be universal.

L 223: a well-mixed test is mentioned but without details. Was it performed with obstacles? as it is more difficult for the scheme? can you comment on this?

The well-mixed tests are based on Bailey (2017) and 3 tests have been considered: 1) synthetic test with zero flow and vertical stress, 2) channel flow DNS data, 3) LES data for the atmospheric boundary layer. In every test, 100,000 particles were uniformly distributed within the domain, and the distribution of particle position was checked after a certain amount of time to ensure that unmixing was not occurring. The well-mixed condition was not tested with obstacles. The difficulty of conducting a well-mixed test with obstacles lies in providing realistic flow and turbulence fields. The text reflects this now.

Equation 20: this analytical solution chosen for the validation is a purely Eulerian solution. Therefore, we do not expect a good agreement with a Lagrangian

solution near the source. Did you compare the near source behavior? What is the Lagrangian time scale (and associated distance from the source?)

We agree with the reviewer that this first test case is not a strong validation case but instead acts as a weak validation of QES-Plume and a stronger verification that the model as implemented is able to capture the analytical solution away from the source furthering our confidence beyond the basic well-mixed tests discussed above. The other cases presented, especially the wind tunnel comparison, offer a strong validation of the model under more realistic conditions.

It is important to note, as pointed out by the reviewer, the chosen Eulerian solution is only valid away from the source. We are unaware of Lagrangian solutions that are strictly valid for times less than the Lagrangian timescale $\tau_L$ (e.g., Taylor, 1921).

To address the need of validation of QES-Plume using a Lagangian approach, we now also compare the trajectory's statistics to the classical dispersion in stationary isotropic turbulence from Pope and Taylor (Pope, 2000, chap. 12, figure 12.10). The results (figure 5) show that QES-plume reproduces both linear and square-root scaling for short distance and long-distance transport, as can be seen in the new figure below:

[Figure]

*Scaling of trajectories in the spanwise direction for homogeneous isotropic turbulence showing the linear spread for $t < \tau_L$. and the square-root spread for $t > \tau_L$. Panel (a) shows the scaling of the standard deviation $\sigma_T(t)$ of the horizontal spread. Panels (b) and (c) show a sample of 200 trajectories in the region of linear spread and square-root spread, respectively, where the black lines represent $\sigma_T$ from Eq. (27).*

Eq. 21-26: because we are in a uniform flow there is no turbulence production by shear. Can you comment on how this turbulent flow is maintained?

In the case of the uniform flow, the friction velocity u* is assumed to be also uniform (horizontally and vertically), and the turbulence is obtained with the

algebraic formulations presented in Eq. 21-26. We modified the text to clarify that u* is an input of the simplified turbulence model.

Eq. 27: this is again an Eulerian solution for which we do not expect a perfect agreement. Why did you not use Taylor's solution which is well suited to Lagrangian validation?
Thank you for the comment. This solution was chosen to include boundary conditions in our evaluation processes (neither the well mixed or elevated plume problems do this). Taylor (1921) does not include boundary conditions and for an exponential correlation function it produces an expression for the concentration field similar to Eq. 20. We are not aware of Lagrangian solutions that analytically account for surface boundary conditions or turbulence profiles the way the solution in Eq. 27 does. If the reviewer has a specific solution in mind that would improve confidence in QES-Plume's evaluation we would be happy to explore it. Still, as noted above, we have added model evaluations based on the scaling arguments of Taylor and Pope.

Figure 5: The concentration gradient at the ground is showing a strange behavior and is very different in the model and the analytical solution. A non-zero concentration gradient is associated with a deposition flux. Can you check and explain this behavior?
We agree that the analytical solution does not seem to respect the zero gradient at the surface. Unfortunately, the number of solutions to the transport equation that account for surface reflection with non-uniform velocity and turbulence fields is very limited. We choose the solution in Eq. 27 because it allowed us to validate the terms with gradient in the GLE.

Figure 8: the normalized concentration profiles are noisier than the measurements. This is unusual and can probably be related to the strong shear discussed in the appendix and linked to the flow wake construction by the diagnostic wind field. Some of the discussion in the appendix should therefore be moved here. As this is an issue, is there a possibility that the flow is smoothed before computing the derivatives for the turbulence?
Thank you for the comment. We added a figure to illustrate the spatial variability of the relative error between the data and the model. With this new figure, we have added some discussion related to the wind from the diagnostic model, especially emphasizing that some of the sharp variations in the profiles are due to the construction of the zones in the mean wind. However, modification of the turbulence model goes beyond the scope of this paper, which focuses on the

Lagrangian model. Improvement of the turbulence model will have to be postponed to a later publication. We have modified the conclusions to reflect the suggestion made by the reviewer.

L 429: it is mentioned a near source and far source behavior as expected in the Lagrangian model. Is it possible to make this more clear by discussing the Lagrangian time in the model and experiment?

Thank you for the comment. Here the near field and far field transports follow the definition by Belcher 2005 specifically for the dispersion in a street network. As far as we are aware, the link between the linear growth and constant width regions in urban network and the Lagrangian timescale has not been clearly established. In addition, the line in question was poorly phrased as it was intended to be interpreted as: "despite poor performance in the first canyon, the rest of the concentration field exhibit behavior comparable to reported in the literature." We have modified the line to avoid the misinterpretation.

Regarding the reviewer's comment relating near source and far source behavior based on the Lagrangian timescale, we have conducted further investigations. However, calculating the Lagrangian timescale from experimental data is difficult, but can be estimated using the ratio of the integral length scale to the turbulence intensity (Tennekes and Lumley, 1972; Davidson et al.,1995). Using the height of the building scaled to the simulation size as estimate of the integral length scale, we get a Lagrangian timescale of $\tau_L$ = 30 - 60 s.

On the other hand, since the code uses the 3D GLE, the Lagrangian timescale is not well defined but can be interpreted as a decorrelation timescale, similarly to section 4.1. Contrary to section 4.1, where the flow was uniform, and the turbulence was isotropic, the complex building array presents some additional challenges to apply the same analysis. Thus, estimating the timescale using the autocorrelation of the local velocity of the particles becomes more difficult and highly variable based on individual particle trajectories. We can approximate the bulk Lagrangian timescale using the ratio of TKE dissipation to variance, which yields $\tau_L \approx$ 50 s. Both estimations of the timescale give the same order of magnitude. Using $\tau_L \approx$ 50 s to identify the section of the trajectories with $\tau_L < 1$ (see figure below), we determine that the 'near source' region corresponds to $1 < x/H < 6$ and $-1.5 < y/H < 1.5$. More exploration of scaling would be interesting but beyond the scope of the current paper. Additionally, as emphasized by Bahlali (2019), trajectories are highly sensitive to mean flow and turbulence estimates. Thus, data from LES would probably be more appropriate by driving QES-Plume to investigate the nature of the scaling. We have added a comment line 404 linking the region of linear growth to $\tau_L < 1$.

[Figure]

*Sample of 1000 trajectories. Blue lines correspond to $\tau_L < 1$. Gray lines correspond to $\tau_L > 1$.*

L 427:"complied": compiled

Thanks for catching the typo.

Figure 9: is there an explanation for the points outside the factor of 10?

From Fig. 9, it can be observed that all the data points outside of the factor 10 class correspond to normalized concentration smaller than 0.005. We can trace these data points to the edge of the plume where both measurement and simulated values present the largest uncertainties. In this case, a small deviation is enough to miss the edge of the plume, leading to large relative errors. We have added some comments to the text.

L464: importance of non-local mixing: this is for the wind tunnel what about the real atmosphere?

The same observation can be made for the real atmosphere. Similar considerations hold for atmospheric flows, where background turbulence levels need to be evaluated from sources such as meteorological measurements or weather prediction models. Thank you for the remark. We have modified the text to reflect this comment.

*References*

Bahlali, M.L., Dupont, E., Carissimo, B., 2019. Atmospheric dispersion using a Lagrangian stochastic approach: Application to an idealized urban area under neutral and stable meteorological conditions. J. Wind Eng. Ind. Aerodyn. 193, 103976. https://doi.org/10.1016/j.jweia.2019.103976.

Bailey, B. N., 2017, Numerical Considerations for Lagrangian Stochastic Dispersion Models: Eliminating Rogue Trajectories, and the Importance of Numerical Accuracy, Boundary-Layer Meteorology, 162, 43–70.

Belcher, S.E., 2005. Mixing and transport in urban areas. Philosophical Transactions Royal Soc Math Phys Eng Sci 363, 2947–2968. https://doi.org/10.1098/rsta.2005.1673

Davidson, M.J., Mylne, K.R., Jones, C.D., Phillips, J.C., Perkins, R.J., Fung, J.C.H., Hunt, J.C.R., 1995. Plume dispersion through large groups of obstacles—A field investigation. Atmos. Environ. 29, 3245–3256. https://doi.org/10.1016/1352-2310(95)00254-v

Ramamurthy, P., Pardyjak, E.R., Klewicki, J.C., 2007. Observations of the Effects of Atmospheric Stability on Turbulence Statistics Deep within an Urban Street Canyon. J. Appl. Meteorol. Clim. 46, 2074–2085. https://doi.org/10.1175/2007jamc1296.1

Taylor, G. I., 1921, 'Diffusion by Continuous Movements', Proc. London Math. Soc. Series 2, 20, 196-212.

Tennekes, Hendrik, and John Leask Lumley. *A first course in turbulence*. MIT press, 1972.

**RC2**: 'Comment on egusphere-2022-1256', Jérémy Bernard, 30 May 2023

This manuscript presents the QES-Plume model, a Lagrangian dispersion model that can be used to quickly estimate particles concentration in fields or in urban areas. The model uses the Generalized Langevin Equations with an implicit time-integration method from Bailey (2017) to alleviate the stiffness problem from the GLEs and eliminate "rogue" trajectories such as described in Yee and Wilson, 2007. The context, need for such model and state-of-the art are very complete and well addressed. The description of the model is very detailed and most of the choices made are well motivated and documented. The model is validated step by step using analytical solutions of simple problems and then compared to data obtained from wind tunnel experiments using arrays of cubes.

The performances are quite good and there is no "rogue" effect observed when applying the model to the arrays of cubes. Most of my comments are minor remarks and propositions to make clearer parts of the article. A single negative comment is that I could not use QES due to computer limitation. It seems that it is necessary to have a NDVIA GPU to use the software for the moment. Hopefully the feature for non-GPU calculation will come soon.

Thanks for the comment. The newest version we pushed to the public git repository can be compile without CUDA on both Linux and MacOS, with support for openMP parallelization of both the wind solver and the Lagrangian model. We have added a comment on the "code availability".

Two main comments:
- that the average concentration error could be added on the spatial representation of the plume on Figure 7 to more easily compare results from the model and observations.
  Thank you for the suggestion. We added figure 10 (section 4.3) showing the spatial distribution of the relative errors. The new figure shows where the largest relative errors are in both lateral and top shear zones behind the buildings, as well as at the edge of the plume. We have added some discussion relating this figure to the discussion in the annex about the turbulence model.

- I would expect having the mean or median RMSE in Table 6 for each class of "Factor X" error in order to illustrate that the error can be relatively important but is probably really low in terms of absolute value.
  Thank you for the comment. We added the RMSE error for each class in Table 6. We have added relative RMSE in Table 5. The discussion linked to the paired scatter plot and Table 6 was extended to reflect comments from the reviewers.

More detail about these two points and all comments can be found in the following extracted annotations:

- « However, the performance of Gaussian models in complex environments is limited » (Margairaz et al., 2022, p. 2)
  -> would be nice to detail further this point (illustrate the "performance limited" and maybe the context and reason when it is limited)
  Thank you for the comment, we have added context about Gaussian plume models, such as the fact that this class of model struggles to capture aspects of the plume such as asymmetry, plume turning, or other critical features especially in urban settings.

- « non-local turbulence mixing coefficient Cnlm » (Margairaz et al., 2022, p. 8)
  -> how is set this coefficient outside the experiment presented in the article?
  The non-local background mixing is not intended to be a universal model for all types of flow, with different atmospheric stability. Background mixing is a model input that needs to be specified for different case and can be obtained from meteorological data such as turbulence measurements, weather models, or adjusted based on stability classes. We have added a comment to the text to reflect that the background mixing constant used is not intended to be universal.

- « Tfinal » (Margairaz et al., 2022, p. 9) on the figure 2 (t < Tfinal condition)
  -> I suppose this is Tfinal but I only have "T in l" in my pdf reader.
  Tfinal is the final time in the while condition. We have added more information in the caption of Fig. 2. We checked that the figure is displayed correctly in Acrobat Reader and Apple Preview. We will work with the copy editor to make sure it does not happen in the final version of the paper.

- « n+1 » (Margairaz et al., 2022, p. 9)
  -> Is n the iterations process? Is that possible to say that somewhere? And maybe to try to add that to the Figure 2?
  The 'n' represents the time step index. Thank you for the suggestion to add it to Fig. 2. We have added where relevant in the workflow figure. We have also modified the text to make clear what 'n' represents.

- « $C_*$ » (Margairaz et al., 2022, p. 12)
  -> not defined (normalized concentration I suppose)
  Yes, C* is the normalized concentration, we have added the definition in the text.

- «flow velocity» (Margairaz et al., 2022, p. 13)
  -> at zs = 70 m? -> Uniform horizontally and vertically?
  We have made sure it is clear in the text that the flow is uniform both horizontally and vertically.

- (Margairaz et al., 2022, p. 16) On bottom figures (e, f, etc.),
  -> x/H values should be added?
  We have added the x/H values in the bottom panel.

- « y/H = 0 » (Margairaz et al., 2022, p. 16)
  -> y/H = 12.5?
  We changed the figure to have 'y' centered on the source, like the other cases.

- (Margairaz et al., 2022, p. 19) Concentration maps
  It would be nice if such map would be used to also illustrate the spatial variability of the error, for example averaging the concentration measured within different zones and comparing them to the model. In my opinion, it would be easier to read and interpret for the reader.
  Thank you for the suggestion, we have added a figure showing the spatial distribution of the relative errors. The manuscript has been adjusted to account for new figure and the related discussion (see new figure below).

[Figure]

*Signed relative error (RE) between the concentration field from QES-Plume and the wind-tunnel measurements where positive values (red markers) represent an overestimation by the model. The panels in the figure are (a) vertical slice at y/H=0 and (b) horizontal slice at z/H = 0.1. The contour lines represent each decade of the concentration form 10 to $10^{-3}$, and the gray contour line $C_* = 10^{-3}$ corresponds to the minimum concentration threshold measurement in the wind-tunnel experiment.*

- « 0.068 » (Margairaz et al., 2022, p. 21)
  -> might be interesting to add the relative value for all RMSE given
  Thank you for the comment. We added the relative RMSE for all the locations. Overall, the mean relative RMSE is 15%.

- « RMSEs » (Margairaz et al., 2022, p. 22)
  -> might it be relevant to also have the relative error?
  Thank you for the comment. We added the relative RMSE for all the locations.

- Table 6
  -> It could be interesting to add the mean (or median) RMSE for each class since the RMSE is probably much lower for factor 10 than factor 2 classes
  We have calculated the RMSE in each class. The RMSE for the factor 2 class is smaller than the RMSE for the other classes. However, we have noticed that in the RMSEs for both factor 5 and factor 10, classes are dominated by 5 points in the profile at $y/H = 0$ and $x/H = 1.5$. If we remove this profile from the calculation of the RMSE, we get values 2 to 5 times smaller. We have added a comment in the text.